RESEARCH

# Population variation in miRNAs and isomiRs and their impact on human immunity to infection

Maxime Rotival[1*], Katherine J. Siddle[2,3], Martin Silvert[1,4], Julien Pothlichet[1,5], Hélène Quach[1,6] and Lluis Quintana-Murci[1,7*] 

* Correspondence: maxime.rotival@
pasteur.fr; quintana@pasteur.fr
[1]Human Evolutionary Genetics Unit,
Institut Pasteur, CNRS UMR 2000,
75015 Paris, France
Full list of author information is
available at the end of the article

## Abstract

**Background:** MicroRNAs (miRNAs) are key regulators of the immune system, yet their variation and contribution to intra- and inter-population differences in immune responses is poorly characterized.

**Results:** We generate 977 miRNA-sequencing profiles from primary monocytes from individuals of African and European ancestry following activation of three TLR pathways (TLR4, TLR1/2, and TLR7/8) or infection with influenza A virus. We find that immune activation leads to important modifications in the miRNA and isomiR repertoire, particularly in response to viral challenges. These changes are much weaker than those observed for protein-coding genes, suggesting stronger selective constraints on the miRNA response to stimulation. This is supported by the limited genetic control of miRNA expression variability (miR-QTLs) and the lower occurrence of gene-environment interactions, in stark contrast with eQTLs that are largely context-dependent. We also detect marked differences in miRNA expression between populations, which are mostly driven by non-genetic factors. On average, miR-QTLs explain approximately 60% of population differences in expression of their cognate miRNAs and, in some cases, evolve adaptively, as shown in Europeans for a miRNA-rich cluster on chromosome 14. Finally, integrating miRNA and mRNA data from the same individuals, we provide evidence that the canonical model of miRNA-driven transcript degradation has a minor impact on miRNA-mRNA correlations, which are, in our setting, mainly driven by co-transcription.

**Conclusion:** Together, our results shed new light onto the factors driving miRNA and isomiR diversity at the population level and constitute a useful resource for evaluating their role in host differences of immunity to infection.

**Keywords:** miRNAs, Isoforms, Population, miR-QTLs, Immunity

## Background

Since their discovery in 1993 [1], microRNAs (miRNAs)—short, evolutionary conserved RNA sequences of ~ 22 nucleotides—have emerged as major regulators of a large variety of developmental and cellular processes such as cell differentiation, proliferation, and homeostasis [2]. There is also increasing evidence that supports their key role in immune responses, with miRNAs such as miR-155 or miR-146a acting to promote and stabilize the inflammatory response [3–7]. Furthermore, numerous studies have reported strong shifts in miRNA expression profiles in response to infectious agents, such as *Mycobacterium tuberculosis* [8, 9], *Salmonella* [10], or influenza A virus [11].

Studies of miRNA abundance across various cell types and tissues have allowed characterizing the extent of genetic regulation of miRNA expression variability, i.e., miRNA expression quantitative trait loci (miR-QTLs), and highlighted the role of genetic variants located in the promoter of the precursor transcript of the miRNA (pri-miRNAs) in shaping inter-individual differences in miRNA expression [8, 12–19]. In the context of immunity, despite increasing evidence that marked population differences exist in the mRNA response to immune challenges [20, 21], the extent to which miRNA responses to infection vary across individuals of different ancestry remains largely unknown.

Fueled by the advent of deep sequencing technologies, growing evidence has emerged that mature miRNAs undergo important post-transcriptional modifications [22–26]. These include nucleotide substitutions (miRNA editing) [27, 28], 3′ adenylation or urydilation by terminal nucleotidyl transferases [29, 30], shortening of their 3′ end by poly(A)-specific ribonuclease [31], and, more rarely, shifts in their 5′ start sites [24]. The diversity of miRNA isoforms (isomiRs) was initially proposed to increase the robustness of miRNA-mediated regulation, by fine-tuning the binding of miRNAs to their targets [32]. Yet, there is now growing support for the notion that miRNA modifications may act as a conserved, additional layer of regulation of their activity [24, 33, 34], as illustrated by the case of miR-222. Upon stimulation with interferon or *Salmonella*, shortening of the 3′ end of miR-222 occurs and leads to a decreased apoptotic action of the miRNA, while maintaining an anti-proliferative effect through the binding of its canonical targets [34, 35]. However, our understanding of the variability of isomiR expression across individuals and populations remains largely incomplete.

Following the canonical model, regulation of gene expression by miRNAs is achieved through the recognition of conserved target sites, which are mostly located in the 3′ UTR of protein-coding transcripts [36–39]. This binding typically results in the repression of the target protein by inducing mRNA deadenylation and degradation or by inhibiting translation [39, 40]. Furthermore, a strong body of evidence highlights the importance of sequence complementarity between the miRNA seed region—located at position 2–7 from the 5′ end of the miRNA [38, 39]—and its target site in determining miRNA-binding. Nonetheless, identifying which mRNAs are actively targeted by a given miRNA remains challenging [41–43]. Previous studies of the regulatory impact of miRNAs on gene expression have reported conflicting results [8, 13, 16, 44], possibly due to difficulties in disentangling the direct effects of miRNAs on mRNA degradation from co-transcription between miRNAs and their targets. In this context, RNA-seq can capture both steady-state gene expression levels, via the analysis of exonic reads, and the dynamic rate of transcription, through the quantification of intronic reads [45]. In

doing so, it offers a unique opportunity to determine the relative contribution of transcription and post-transcriptional regulation by miRNAs to gene expression variability.

In this study, we provide a comprehensive resource of genome-wide sequence-based miRNA diversity from primary human monocytes, both at the basal state and upon cellular treatment with four immune stimuli, originating from 200 individuals of African and European descent (100 individuals from each ancestry, Fig. 1). Leveraging the information obtained from 977 small RNA-sequencing profiles, together with whole-genome genotyping and exome sequencing data as well as mRNA-sequencing data from the same individuals, we define the levels of miRNA and isomiR diversity across individuals and populations, explore the genetic sources of miRNA expression

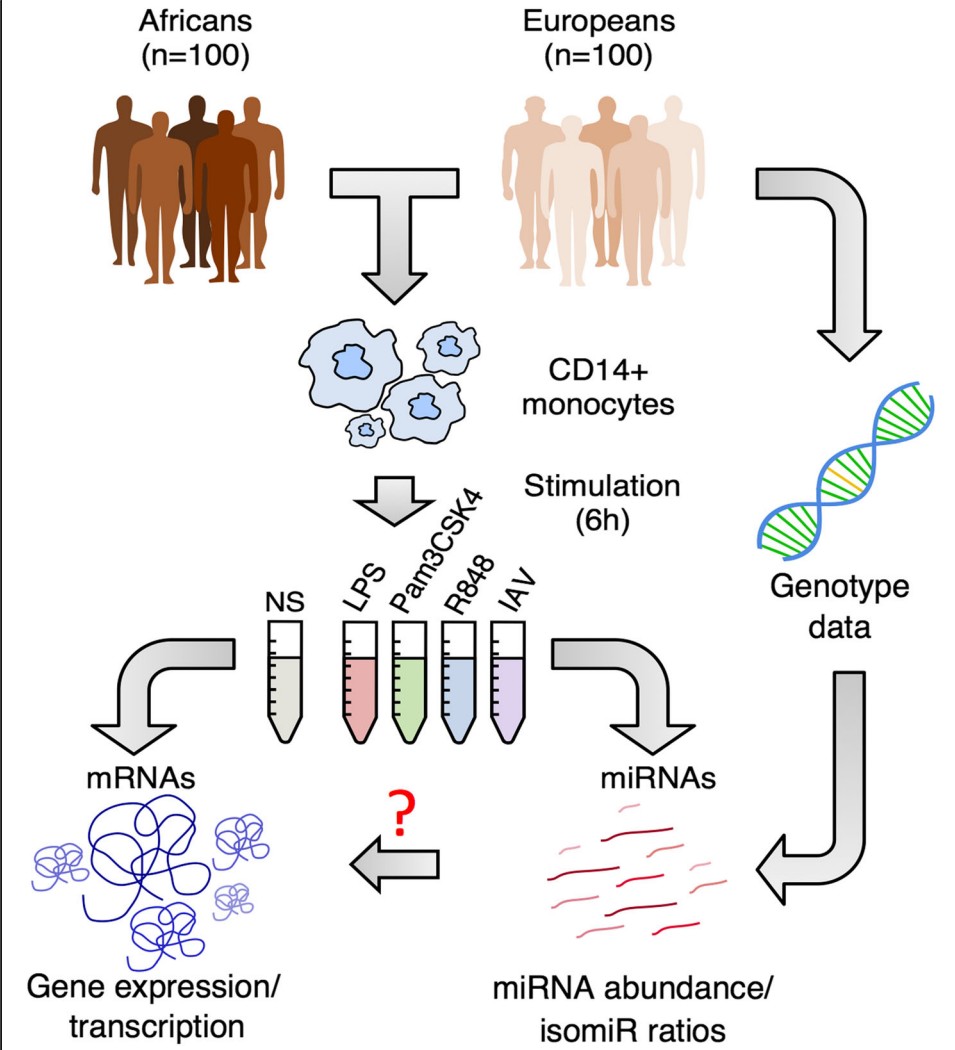

**Fig. 1** Population variation in miRNA response to immune activation. We stimulated monocytes from 200 healthy individuals using 3 TLR ligands as well as a strain of influenza A virus (IAV). For each individual, RNAs were extracted after 6 h of stimulation. We sequenced small RNAs, from both stimulated cells and non-stimulated cells (NS) (this study). The integration of miRNA sequencing data with genetic data, obtained through whole genome genotyping, whole-exome sequencing, and imputation [21], allowed us to assess the genetic bases of population differences in miRNA responses to stimulation, both quantitatively (miRNA abundance) and qualitatively (isomiR ratios). Furthermore, the availability of mRNA sequencing data from the same individuals and experimental conditions allows quantifying both total gene expression levels (exonic reads) and transcription rate (intronic reads derived from nascent mRNAs)

variability and miRNA-environment interactions, evaluate the effects of immune challenges upon miRNA and isomiR expression dynamics, and quantify the relative impact of transcription and miRNA-mediated degradation on gene expression variability.

## Results

### The landscape of miRNA and isomiR expression in human monocytes

We generated 977 small RNA sequencing profiles, in resting and activated cells, from 200 healthy individuals of African and European ancestry. Activation was performed for 6 h with three different Toll-like receptor (TLR) ligands (LPS, $Pam_3CSK_4$, and R848 activating TLR4, TLR1/2, and TLR7/8 pathways, respectively) and a live strain of influenza A virus (IAV, Fig. 1). Small RNAs were separated from mRNAs and sequenced at a mean depth of 12.4 million reads per sample (see the "Materials and methods" section; Additional file 1: Fig. S1a-c). After excluding reads outside the 18–26 nt range and low-quality samples (Additional file 1: Fig. S1d,e), we obtained an average of ~5 million reads aligned to miRNAs. To correct for cross-mapping artifacts between miRNAs, multiply-mapped reads were assigned to each possible locus using an Expectation-Maximization strategy [28]. Library size was normalized across samples, and miRNAs with an average of < 1 read per million miRNA-mapped reads (RPM) were discarded. This yielded a final set of 736 loci, encoding for 658 distinct miRNAs (Additional file 2: Table S1).

Focusing on unique sequences, we identified 23,447 putative isomiRs, the vast majority (90%) of which were lowly abundant (< 1 RPM; 14,277 isomiRs) or extremely rare (< 1% of the reads of the associated miRNA; 6811 isomiRs). Focusing on the remaining 2359 unique miRNA sequences (corresponding to 492 loci encoding 451 distinct miRNAs, Additional file 2: Table S1), we found that 86% of miRNAs expressed one or more isomiR(s) beside the canonical form, with a single miRNA expressing up to 8 frequent isomiRs (> 5% reads) (Fig. 2a and Additional file 1: Fig. S2a,b). For more than 57% of miRNAs, the canonical isomiR accounted for less than half of the copies of the miRNA (Fig. 2b). Among the 311 miRNAs where the canonical isomiR was in minority (< 50% of the reads), 25% had a seed sequence that differed from the canonical isomiR in more than 20% of their copies (Fig. 2c).

### Dissecting the mechanisms underlying miRNA isoform diversity

To dissect the processes leading to the high isomiR diversity observed, we pooled miRNAs from each condition and quantified each type of miRNA modification separately (i.e., shifts in start/end site, non-template additions [NTA], and substitutions were quantified independently, and isomiRs with > 1 modification were counted multiple times, see the "Materials and methods" section). We found that the overall frequency of miRNA modifications was virtually unchanged by stimulation (Wilcoxon $p > 0.05$, for all stimuli relative to basal state). Shifts in the 3′ end site of miRNAs were the most frequent type of modification. In total, ~ 80% of miRNAs presented a shift of their 3′ end site in > 5% of the reads, even after exclusion of non-template additions (Additional file 1: Fig. S2c,d; ~ 87% including NTA), consistent with previous results [9]. Conversely, only ~ 31% of miRNAs presented a frequent shift of their 5′ start site (> 5% of reads, Additional file 1: Fig. S2e), reflecting strong constraints on the miRNA seed.

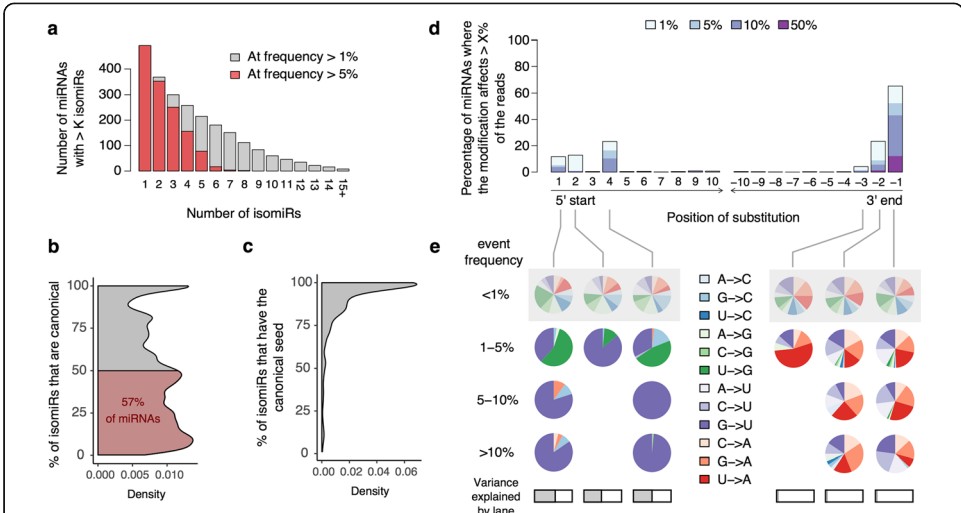

**Fig. 2** The landscape of miRNA diversity in human primary monocytes. **a** For each possible number of isomiRs *K*, number of miRNAs with more than *K* isomiRs at a frequency of > 5% or > 1%. **b** Distribution of the percentage of canonical isomiRs among all miRNAs; the area where the canonical isomiRs account for less than 50% of the reads is highlighted in red. **c** Distribution of the percentage of isomiRs with canonical seed among all miRNAs. **d** Distribution of edited nucleotides along miRNAs. For each nucleotide position, counting either from the 5′ start site (position 1 to 10) or from the 3′ end site (positions − 10 to − 1), we report the percentage of miRNAs that present an editing event accounting for $\geq$ 1% of reads (light blue). Similarly, we quantified the fraction of miRNAs where the editing event accounts for > 5% (blue), > 10% (indigo), or > 50% (deep purple) of the reads. **e** Frequency of each type of substitution, according to the percentage of miRNA reads that are edited. Results are shown for positions where events were detected in > 1% of miRNAs. At each position, distribution of substitution types among low frequency editing events (< 1%), which we expect to be enriched in false positives, is provided as a reference (gray shadow). At each position, a horizontal gray bar indicates the variance in sample editing levels that is accounted for by the sequencing lane

Focusing on nucleotide substitutions, we found a strong enrichment of substitutions at 3′ end of the miRNA (binomial $p < 3.7 \times 10^{-38}$, Fig. 2d, e). This enrichment recapitulated known patterns of 3′ terminal uridylation and adenylation [23] and was unaffected by the stimulation state (Wilcoxon $p > 0.05$, for all stimuli relative to basal state). We also detected a strong enrichment of substitutions at the 5′ end, as well as at the seed-altering positions 2 and 4 of the miRNA (binomial $p < 2.2 \times 10^{-5}$). All three positions (i.e., nucleotides 1, 2, and 4) presented a strong bias (binomial $p < 9.8 \times 10^{-31}$) toward G- > U substitutions, as well as low frequency U- > G changes at positions 1 and 4 (binomial $p < 0.003$). While the frequency of terminal substitutions was stable across sequencing batches ($R^2 < 0.1\%$), substitutions at positions 1, 2, and 4 depended on the sequencing lane ($R^2 > 48\%$), suggesting a technical bias. These substitutions were thus not considered for isomiR definition. IsomiR abundances were recomputed merging all isomiRs that differed by non-terminal substitutions and considering only isomiRs that result from shifts in the start or end of the miRNA or 3′ terminal uridylation and adenylation. After removing spurious isomiRs, the final dataset consisted of 2049 frequent isomiRs across 435 miRNAs (Additional file 2: Table S1).

We compared the frequency of miRNA modifications across both arms of the pre-miRNA hairpin (Additional file 2: Table S1). We observed a stronger degree of 3′ terminal uridylation at 3p miRNAs (+ 12% of uridylated miRNAs on 3p arm compared to 5p; Wilcoxon $p < 2.8 \times 10^{-11}$, Additional file 1: Fig. S2f), consistent with the reported

role of uridyl-transferases in pre-miRNA maturation [26, 46, 47]. This increased uridy-lation was not associated to a higher rate of 3′ extensions among miRNAs located on the 3p arm (Wilcoxon $p = 0.47$), due to a higher rate of template extensions among 5p miRNAs ($+ 7.6\%$ on 5p arm compared to 3p; Wilcoxon $p < 0.006$, Additional file 1: Fig. S2g,h). Finally, we detected a higher usage of non-canonical, downstream start sites among 3p miRNAs ($+ 3\%$ compared to 5p miRNAs; Wilcoxon $p < 0.003$), consistent with a regulation of isomiR variability through the tuning of DICER positioning on the pre-miRNA [48]. Overall, the high variability of isomiRs detected highlights the com-plexity of the landscape of miRNA modifications in human primary monocytes.

### Marked effects of immune challenges upon miRNA and isomiR expression

Principal component (PC) analysis of miRNA abundances revealed a clear separation by stimulation conditions (Fig. 3a), after adjusting miRNA and isomiR expression for batch effects (date of experiment, date of library preparation and sequencing lane) and technical confounders (GC content and mean read length of the sample). PC1 opposed TLR-activated from IAV-infected samples, while PC2 captured the shared effect of all immune stimuli on gene expression. The variance explained by these PCs (i.e., a total of 16.7%) indicates that the effect of immune activation on miRNA expression is much weaker than that observed for protein-coding mRNAs measured on the same individual samples [21], where stimulation explained ∼ 69% of expression variability. In contrast with patterns at the mRNA level, we noticed significant shifts between populations on both PCs (PC1, $t$ test $p < 1.0 \times 10^{-79}$; PC2, $t$ test $p < 1.2 \times 10^{-11}$), possibly reflecting differences in the intensity of miRNA responses to immune stimuli between individuals of African- and European-ancestry.

At FDR < 1%, we identified 340 miRNAs that presented differential expression upon stimulation ("DE miRNAs", 30 with $\log_2 FC > 1$), 233 of which were upregulated in at least one condition (58–74% per condition; Fig. 3b and Additional file 3: Table S2). Notably, DE miRNAs were observed across all levels on gene expression, with a slightly higher proportion among genes with > 10 RPM (OR = 2.0, Fisher's $p < 4.4 \times 10^{-16}$, Additional file 1: Fig. S3a,b). Using a likelihood-based model selection framework [49] (Fig. 3c), we estimated that 90% of DE miRNAs respond in a stimulus-dependent manner. The three most frequent patterns of miRNA responses were (i) a TLR-specific response ($N = 65$, 19% of DE miRNAs), as in the case of the NF-κB inhibitors miR-9-5p (Fig. 3d) and miR-155-5p; (ii) a viral-stimuli specific response (R848 and IAV, $N = 55$, 16% of DE miRNAs), such as miR-3614-5p recently involved in Crohn's disease susceptibility [50] (Fig. 3e); and (iii) an IAV-specific response ($N = 78$, 23% of DE miRNAs), as attested by the pro-inflammatory mir-429 or the TRIM22 repressor mir-215-5p (Fig. 3f).

Focusing on how immune activation altered isomiR ratios, IAV infection clearly had the strongest impact (PC1, 4.7% of variance explained) followed by TLR7/8 activation (PC2, 2.6% of variance explained), with TLR4 and TLR1/2 activation showing a limited impact (Additional file 1: Fig. S3c). A total of 316 miRNAs changed their isomiR ratios upon stimulation (Additional file 3: Table S2). Among these, the ratio of the canonical form was found to be affected for 212 miRNAs (67%), a ratio that decreases in 56 to 70% of the cases (Fig. 3g). For a majority of miRNAs, changes in isomiR ratios were of

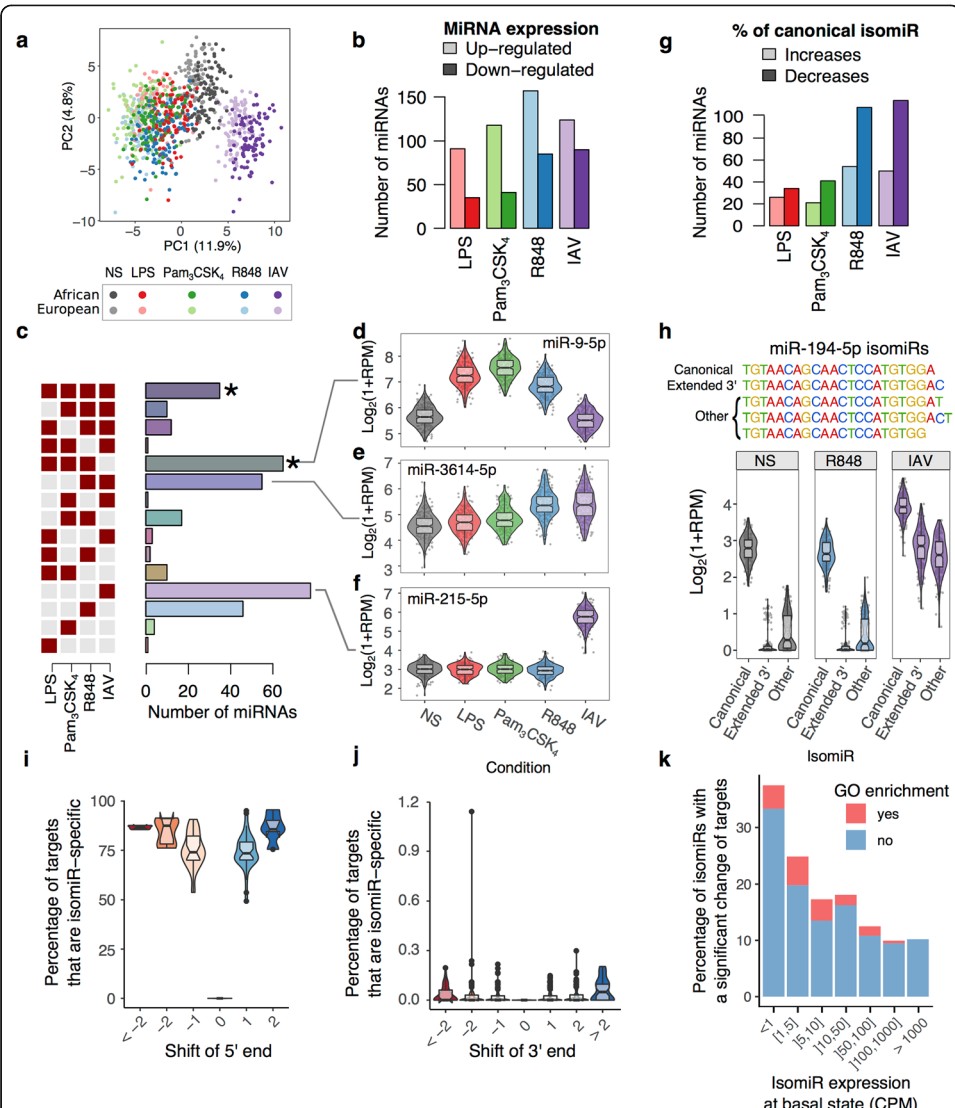

**Fig. 3** Stimulus-specific miRNA responses to immune activation. **a** PCA of log transformed miRNA abundances. Each dot represents a sample, colored according to the experimental condition (gray – non-stimulated, red – LPS, green – Pam₃CSK₄, blue – R848, purple – IAV). The same color code for conditions is used throughout the manuscript, and light and dark shades indicate European and African ancestry, respectively. **b** For each condition, number of DE miRNAs that are either up- (light shade) or down-regulated (dark shade). **c** Number of miRNAs that are differentially expressed (compared to NS) in a single condition or a combination of immune stimulations (*binomial $p < 0.001$, significance of overlap between stimuli). **d–f** Examples of DE miRNAs for the three most frequent patterns of differential expression across stimuli: **d** miR-9-5p, exhibiting a TLR-specific response; **e** miR-3614-5p, responding specifically to viral stimuli; **f** miR-215-5p, showing an IAV-specific response. **g** For each condition, number of miRNAs where the canonical isomiR is either up- (light shade) or down-regulated (dark shade). **h** Example of isomiR-level response to IAV for miR-194-5p. miR-194-5p expresses 5 isomiRs that differ in their 3' end. Violin plots show the expression of miR-194-5p isomiRs at the basal state, after R848 stimulation (as an example of a TLR-ligand), and IAV. The 3 least frequent isomiRs are grouped for clarity. **i** Distribution of the percentage of targets from frequent isomiRs (> 1% of miRNA reads, RPM > 1) that differ from those of the canonical isomiR, as a function of shifts in the 5' start site. Only isomiRs that possess the canonical end site were considered here. **j** Distribution of the percentage of targets from frequent isomiRs that differ from those of the canonical isomiR, as a function of shift in 3' end site. Only isomiRs that possess the canonical start site were considered here. **k** Percentage of common non-canonical isomiRs for which > 1% of their targets differ from those of the canonical isomiR, across various bins of isomiR expression. For each bin, isomiRs whose targets are enriched for at least one GO category are shown in red

moderate intensity, with a single isomiR changing its ratio by more than 5% in only 5–11% of miRNAs (Additional file 1: Fig. S3d). Notable exceptions included mir-155-5p, shifting toward extended 3′ isomiRs upon stimulation (13–33% increase in longer iso-miRs, Wilcoxon $p < 2.7 \times 10^{-68}$), and miR-194-5p, showing an IAV-specific shift toward a 3′-extended isomiR (17% increase in longer isomiR upon infection, Wilcoxon $p < 7.4 \times 10^{-97}$, Fig. 3h). Overall, changes in isomiRs were most frequent following treatment with viral ligands (R848 and IAV), with 36% of isomiRs changes being shared between these two stimuli (Additional file 1: Fig. S3e).

We next investigated whether this observation could be explained by specific mechanisms of miRNA modifications (Additional file 3: Table S2). Despite high stability between conditions in the distribution of 5′ and 3′ shifts of miRNAs, paired analyses—comparing each miRNA between the basal and the stimulated state—revealed a recurrent trend toward 3′ shortening of miRNAs upon R848 and IAV conditions (0.5% and 0.9% increase in frequency of 3′ end shortening of miRNAs, respectively; paired Wilcoxon $p_{\text{adj}} < 3.9 \times 10^{-14}$), reflecting a global increase in the rate of 3′ shortening (Additional file 1: Fig. S3f). Changes of the miRNA reading frame accounted for > 84% of such patterns, while the remainder was mostly attributable to a reduction of 3′ uridylation (16% and 10% for R848 and IAV, respectively; paired Wilcoxon $p_{\text{adj}} < 1.8 \times 10^{-10}$, Additional file 1: Fig. S3g,h). These results collectively highlight significant shifts in miRNAs expression upon immune stimulation and reveal a high rate of isomiR modifications in response to viral stimuli.

### Functional impact of miRNA modifications upon immune stimulation

We then explored the extent to which miRNA modifications, at the basal state or upon stimulation, may alter their targets and thus, potentially, their biological function. To do so, we used miRanda [51] to predict the targets of frequent non-canonical isomiRs (> 1% of the reads and > 1RPM) and compared them to those of the canonical isomiR (Additional file 3: Table S2). We found that 43% of frequent non-canonical isomiRs are associated with a gain/loss of miRNA targets, including 19% that changed > 1% of predicted targets. Consistent with the importance of the miRNA seed in target prediction algorithms, shifts of the 5′ boundary of miRNAs were nearly systematically predicted to alter between 46% and 96% of their targets (Fig. 3i). Conversely, shifts of the 3′ boundary were predicted to have minor effects, typically affecting < 0.1% of their targets (Fig. 3j).

Notably, highly expressed isomiRs were less likely to alter miRNA targets than low expressed isomiRs (Fig. 3k), suggesting selective constraints limiting the expression of isomiRs that had significant downstream effects. Despite this general trend, for ∼ 10% of isomiRs expressed at > 1000 RPM, > 10% of the predicted targets differed from those of the canonical isomiR, supporting a sizeable impact of isomiR variation on the target repertoire of miRNAs. Furthermore, we identified 26 miRNAs (6.8% of tested miRNAs) for which targets that are unique to either the canonical or the non-canonical isomiR are enriched for specific biological functions (Additional file 3: Table S2). For 10 of these miRNAs, stimulation altered the expression of isomiRs with non-canonical targets. For example, a 5′-shifted isomiR of miR-449c-5p is downregulated upon R848 and IAV challenges ($\Delta_{\text{isomiR-ratio}} > 6.9\%$, Wilcoxon $p < 3.4 \times 10^{-6}$) and is associated with

the loss of 39 targets involved in homophilic cell adhesion (GO:0007156, OR = 2.5, Fisher's $p < 3.3 \times 10^{-6}$), consistent with previous experimental observations [52]. Similarly, changes in the 5′ end of miR-6503-3p, in response to viral stimuli, lead to a reduced proportion of canonical isomiRs ($|\Delta_{isomiR\text{-}ratio}| = 0.8\text{–}1.1\%$, Wilcoxon $p < 4.6 \times 10^{-9}$). Interestingly, all non-canonical isomiRs of miR-6503-3p converge to the regulation of type-I interferon genes (OR > 11.6, Fisher's $p < 1.2 \times 10^{-10}$), despite presenting 3 different seed sequences. This suggests a role of miR-6503-3p isomiRs in the regulation of type-I interferon antiviral response.

Altogether, these results indicate that although > 80% of isomiRs share their targets with the canonical form, shifts of the miRNA start site may occasionally lead to repurposing miRNA function by altering the biological pathways they target.

### Strong selective constraints limit miRNA expression variability

To assess the extent to which miRNA expression variability is under genetic control, we focused on the 598 miRNAs associated to a unique genomic location and searched for genetic variants associated with changes in miRNA abundances within a 1 Mb window around each miRNA (miRNA Quantitative Trait Loci, or miR-QTLs). At 5% FDR, we identified 122 miRNAs associated with at least one miR-QTL (Additional file 4: Table S3), corresponding to ~ 20% of the tested miRNAs. Interestingly, this proportion is lower than that observed for mRNAs of protein-coding genes or long non-coding RNAs (with 31% and 33% of the latter presenting an eQTL, respectively, Fisher's $p < 1.2 \times 10^{-7}$). However, we found a comparable proportion of eQTLs among transcription factors and loss-of-function intolerant genes (25% and 22% of eQTLs, respectively, Fig. 4a). Furthermore, we observed a decreased proportion of miR-QTLs among highly expressed miRNAs (Additional file 1: Fig. S4a,b). Interestingly, miRNAs with conserved promoters (mean phastCons > 20%) were depleted in miR-QTLs with respect to miRNAs with less conserved promoters (OR = 0.54, Fisher's $p < 0.008$). In addition, miR-QTLs of miRNAs with a conserved promoter were located on average further away from the transcription start site (TSS; + 3.5 kb, Wilcoxon $p < 0.03$, Additional file 1: Fig. S4c). Collectively, these observations indicate strong selective constraints that limit miRNA expression variability

### miR-QTLs are largely shared across immune stimuli

When comparing the occurrence of miR-QTLs across stimuli, we found that 85% of all miR-QTLs were shared across all experimental conditions (Fig. 4b and Additional file 4: Table S3), with a minority (N = 18, 15%) displaying condition-dependent effects and only 5.7% being specific to one condition. This observation is in stark contrast with eQTLs of protein-coding genes, where 53% of them displayed condition-dependent effects (Fig. 4b). However, the proportion of miR-QTLs and eQTLs that are specific of one condition was quite similar (5.7 and 6.2%, respectively; Fig. 4b). We then assessed whether the lower frequency of condition-dependent miR-QTLs, relative to eQTLs, could be attributed to the higher stability of miRNA expression upon stimulation. To do so, we modeled the probability of miR- and eQTLs to be condition-dependent as a function of the maximal absolute $\log_2$ fold change in response to stimulation (Additional file 1: Fig. S4d,e). While condition-dependent QTLs were found to be more

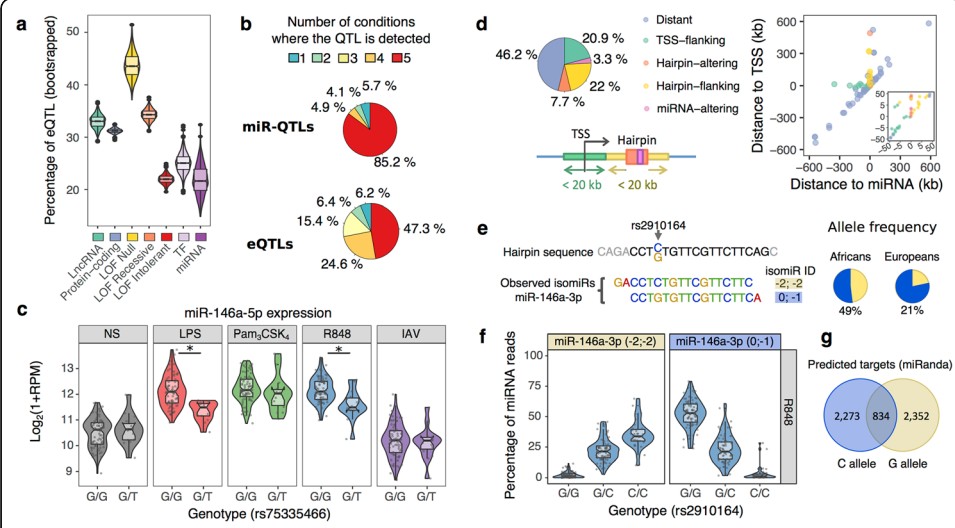

**Fig. 4** Genetic basis of miRNA expression upon immune activation. **a** Percentage of genes and miRNAs under genetic control, across various functional classes. For each gene class, 1000 bootstrap resamples were performed and the resulting distribution is shown as a boxplot. **b** Sharing of miR-QTLs and eQTLs across conditions. For the 122 miR-QTLs and 3802 eQTLs, number of conditions where a QTL is detected. **c** Example of an African-specific response miR-QTL. The rs75335466-T allele is associated with reduced expression of miR-146a-5p specifically upon stimulation by LPS and R848 (* *t* test *p*-value < 0.001). For clarity, data is shown only for African-ancestry individuals. **d** Localization of miR-QTLs. Left: Frequency of miR-QTLs that either overlap the mature miRNA (*miRNA-altering*, pink) or its hairpin (*hairpin-altering*, orange), or are located < 20 kb away from the miRNA hairpin (*hairpin-flanking*, yellow) or TSS (*TSS-flanking*, green). Remaining miR-QTLs are annotated as *Distant* (blue). Right: Distance of mir-QTLs from the mature miRNAs (*x*-axis) and its associated TSS (*y*-axis). Each miR-QTL is shown as a separate dot, colored according to its localization. Negative distance indicates that the miR-QTL is located upstream of the miRNA and/or TSS. Close-up view is shown for miR-QTLs located < 50 kb from the miRNA or TSS. (*E–G*) Impact of rs2910164 variant on miR-146a-3p isomiRs. **e** Genomic context and frequency of the rs2910164 variant. The rs2910164 C/G variant is shown with its neighboring hairpin sequence. Sequence of the canonical miRNA is displayed in black. The 2 most frequent isomiRs of miR-146a-3p are displayed below and denoted as (− 2; − 2), and (0; − 1) based on the coordinates of their start/end site relative to the canonical miRNA sequence. Note that the C/G substitution is not considered for the quantification of (− 2; − 2) and (0; − 1) isomiRs. Frequency of C/G alleles in our sample is shown in African- and European- ancestry individuals separately. **f** isomiR-QTL of miR-146a-3p. Ratios of (− 2;-2) and (0; − 1) isomiRs are shown for each genotype, in the R848 condition where the isomiR-QTL is the strongest. For clarity, other isomiRs are not displayed. **g** Overlap of miRNA targets predicted by miRNA for each possible isomiR

frequent among protein-coding genes and miRNAs that respond to stimulation (OR = 1.26 per $\log_2$FC of gene expression, likelihood ratio p value < $1.7 \times 10^{-7}$), miRNAs remained less likely to harbor condition-dependent QTLs relative to protein-coding genes (OR = 0.2, likelihood ratio *p* value < $5.7 \times 10^{-10}$), even after considering their weaker response to stimulation.

Among condition-dependent miR-QTLs, we detected 4 response miR-QTLs, i.e., genetic variants that manifest their effects on miRNA abundance only in the presence of immune stimulation ($p_{\text{interaction}}$ < 0.001, corresponding to 5% FDR, and not detected in the non-stimulated state). For example, the African-specific rs75335466 has a derived allele (derived allele frequency (DAF) = 7.5% in African-ancestry individuals) that is associated, upon stimulation of TLR4 and TLR1/2 pathways, to a reduced upregulation of the dominant arm of miR-146a (miR-146a-5p, $p_{\text{interaction}}$ < $5.6 \times 10^{-4}$, Fig. 4c), which acts as an inhibitor of TRAF6 and IRAK1 [53]. Overall, these results show that miR-QTLs display a

reduced sensitivity to immune stimulation, with respect to eQTLs, an observation that cannot be accounted by the weaker miRNA response to stimulation.

### Genetic control of miRNAs is largely independent from that of protein-coding genes

Given the reported role of enhancers in leading responses to immune stimulation [54], we hypothesized that the limited condition-specificity of miR-QTLs might be driven by a different regulatory architecture from that of eQTLs. We first sought to characterize the regulatory elements underlying miRNA expression. We found that 54% of miR-QTLs were located < 20 kb from either the TSS of the pri-miRNA they regulate or the pre-miRNA hairpin that contains the mature miRNA (Fig. 4d). Furthermore, we observed a strong over-representation of both promoters (OR = 17, Fisher's $p < 4.5 \times 10^{-12}$) and enhancers (OR = 3.6, Fisher's $p < 5.9 \times 10^{-5}$) among miR-QTLs. Yet, the percentage of miR-QTLs falling into promoters or enhancers did not differ significantly from that observed for eQTLs (Fisher's exact test; $p = 0.25$ for promoters and $p = 0.88$ for enhancers; Additional file 1: Fig. S4f).

We then focused on the 352 miRNAs that are located in introns of protein-coding genes expressed in our setting (intronic miRNAs), to assess how their genetic control overlaps with that of their host genes. Although 81 miRNAs were located within a gene whose expression is under genetic control, the corresponding eQTL had a significant impact on miRNA expression for only a quarter of them ($N = 20$, 1% FDR). Likewise, of the 64 miR-QTLs that alter the expression of intronic miRNAs, only 6 were in high LD ($r^2 > 0.8$) with an eQTL of their corresponding host gene. Furthermore, for 5 of the latter, likelihood-based causality inference [55] supported an independent effect of genetics on miRNA and host gene expression. Only miR-147b had a miR-QTL whose effect on miRNA expression was predicted to be mediated by the regulation of its host gene *AATK*. Overall, these results reveal that despite similar enrichments in promoter and enhancer regions, the genetic control of miRNAs is largely independent from that of protein-coding genes.

### Limited genetic control of isomiR diversity

We then searched for genetic variants that alter isomiR ratios (isomiR-QTLs). Only 25 isomiRs were associated with at least one isomiR-QTL, involving 13 miRNAs (Additional file 4: Table S3), 84% of these being shared across conditions. Note that because we did not consider non-terminal substitutions in our definition of isomiRs, these numbers do not take into consideration genetic variants that directly alter the miRNA sequence, unless they also alter the start/end site of the miRNA. An interesting case of isomiR-QTL is provided by the rs2910164 variant (DAF: 49% in African and 21% in European ancestry groups), which disrupts the seed of the passenger arm of miR-146a (miR-146a-3p, Fig. 4e). The derived allele of rs2910164 (G) is associated with both an increase in expression of miR-146a-3p ($t$ test, $|\beta_{\text{miR-QTL}}| > 0.31$, $p < 3.1 \times 10^{-7}$) and a shift of both the start and end sites of the mature miRNA ($t$ test, $|\beta_{\text{isomiR-QTL}}| > 0.15$, $p < 2.1 \times 10^{-9}$, Fig. 4f). This shift leads to a complete redefining of the miR-146a-3p targets, with 2273 predicted targets being lost (73%) and 2352 novel targets being gained (Fig. 4g). Interestingly, the rs2910164-G allele is associated with increased risk of allergic rhinitis ($p < 1.9 \times 10^{-13}$) and asthma ($p < 6.2 \times 10^{-9}$) in the GWAS Atlas [56].

Despite the limited genetic control of isomiR diversity, these results highlight how genetic variants altering isomiR ratios can lead to a profound rewiring of targets from key immune regulators.

### Marked differences in miRNA expression related to population ancestry

We subsequently explored the extent to which miRNA responses to stimulation differ between individuals of African and European ancestry. We identified a total of 351 miRNAs whose transcriptional profiles differed between populations in at least one experimental condition, either in abundance ("pop-DE-miR", $N = 244$, including 141 with $|\log_2 FC| > 0.2$, Fig. 5a and Additional file 5: Table S4), or in isomiR ratios ("pop-DE-isomiR", $N = 188$, including 148 with $\Delta_{\text{isomiR-ratio}} > 1\%$, Fig. 5b and Additional file 5: Table S4), with 81 miRNAs differing in both expression and isomiRs. We found that at the basal state population differences in expression of miRNAs were similar in magnitude to those of protein-coding genes (17% of miRNAs and 12% of protein-coding

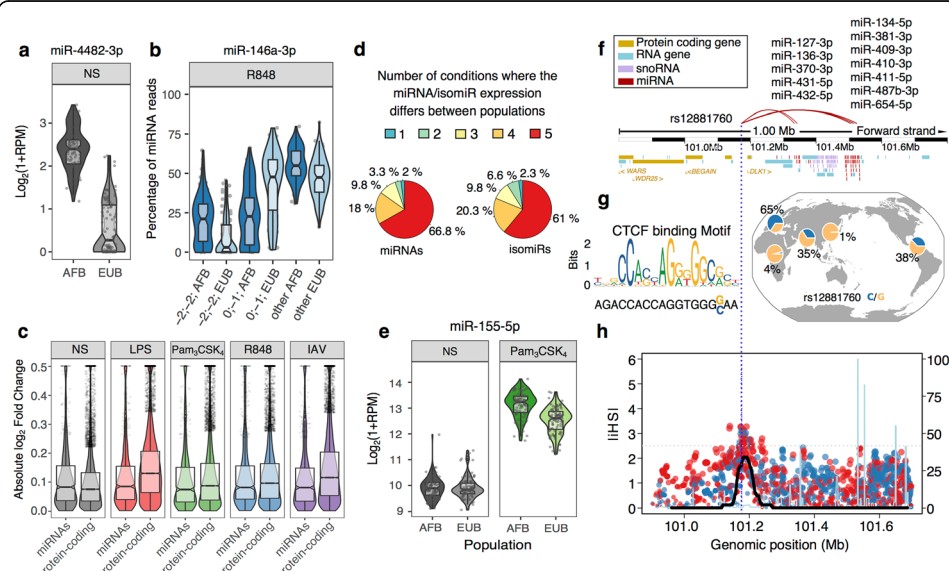

**Fig. 5** Population differences in miRNA expression. **a** Example of a miRNA (miR-4482-3p) differentially expressed between populations. Expression of miR-4482-3p is shown separately for African- (AFB) and European- (EUB) ancestry individuals. **b** IsomiRs of miR-146a-3p are differentially expressed between populations. For each population and isomiR, isomiR ratios are shown in the R848 condition where the difference is the strongest. All isomiRs with < 1 RPM on average, are pooled and annotated as *other*. **c** Amplitude of population differences in expression of miRNAs and protein-coding genes, at basal state and upon stimulation. **d** Sharing of pop-DE-miRs and pop-DE-isomiRs across conditions. For the 244 pop-DE-miRs and 188 pop-DE-isomiRs, number of conditions where we observed a difference between populations. **e** Expression of miR-155-5p is differential between African- and European-ancestry individuals, specifically in TLR-stimulated conditions. For simplicity, only Pam3CSK4 condition is shown. **f–h** Signatures of positive selection targeting the miR-QTL hotspot rs12881760. **f** Genomic context of the miR-QTL hotspot, displaying protein coding genes (yellow), RNA genes (cyan), snoRNAs (purple), and miRNAs (red) in a 1 Mb-window around the locus. Red lines link the miR-QTL to its target miRNAs, the name of which are indicated above. **g** Impact of the rs12881760 variant on a CTCF motif, and worldwide frequency of the motif-disrupting C allele. **h** Signatures of positive selection at the rs12881760 locus. |iHS| are displayed for all SNPs with MAF > 5% in Europeans, and dots are colored according to the sign of the iHS statistic (red, positive; blue, negative). The black line indicates the percentage of outliers (|iHS| > 2.5) on a sliding window of 100 consecutive SNPs with a MAF > 5% (right axis). Recombination rate is overlaid in light blue and normalized to the maximum recombination rate in the region (peak, 152 cM/Mb)

genes display a $|\log_2 FC| > 0.2$ between populations, Wilcoxon $p = 0.08$, Fig. 5c). Upon stimulation, however, protein-coding genes displayed a marked increase in population differences (17–26% with a $|\log_2 FC| > 0.2$ between populations, Wilcoxon $p < 2.7 \times 10^{-27}$), while population differences in miRNA expression remained rather stable (14–18% with a $|\log_2 FC| > 0.2$ between populations, Wilcoxon $p > 0.26$, Fig. 5c).

Population differences in miRNA expression and isomiRs were largely shared across experimental conditions, with 67% of pop-DE-miRs and 61% of pop-DE-isomiRs being shared across all experimental conditions (Fig. 5d). Yet, we identified 9 miRNAs that displayed population differences only upon stimulation (Additional file 5: Table S4), including key immune modulators such as the pro-inflammatory miR-155-5p, which showed marked population differences upon TLR1/2 stimulation (Pam$_3$CSK$_4$; $p_{\text{interaction}} < 1.0 \times 10^{-9}$, Fig. 5e). Looking at the rate of miRNA modifications, we found that the 3′-end shortening of miRNAs located on the 3p arm was more frequent in individuals of African ancestry with respect to those of European-ancestry (Wilcoxon $p < 3.2 \times 10^{-6}$ at basal state, Additional file 5: Table S4). Furthermore, individuals of African ancestry presented, upon stimulation, an increased rate of 3′ adenylation, regardless of the arm where the miRNA is located (Wilcoxon $p < 4.5 \times 10^{-5}$), partially compensating the detected shortening of miRNAs located on the 3p arm.

### Sources of ancestry differences in miRNA and isomiR responses

We next searched for the sources of population differences in miRNA expression and found a significant enrichment of pop-DE-miRs and -isomiRs in miRNAs whose expression is under genetic control (i.e., miR- or isomiR-QTL; OR > 1.7, Fisher's $p < 1.1 \times 10^{-2}$). By computing the fraction of population differences in miRNA expression that is attributable to genetic factors, we estimated that, among the 57 pop-DE-miRs with a miR-QTL (23% of popDE-miR), genetics accounted for ~ 60% of population differences on average. Across all miR-QTLs, the strongest differences in frequency between individuals of African and European ancestry were observed at the variant rs12881760 on chromosome 14. This variant is associated with the expression of 12 miRNAs that are located in a cluster of 148 small RNAs spanning over 250 kb (Fig. 5f). The derived allele (C) disrupts a CTCF binding site located ~ 200 kb upstream of the small RNA cluster and is associated with a lower platelet mass in the GWAS Atlas ($p < 6.5 \times 10^{-48}$) [56]. Interestingly, the C allele is found at high frequency in European-descent populations (e.g., up to 72% in Iberians) and rare in Africans and East Asians (< 4%, Fig. 5g). Moreover, it harbors a strong signature of positive selection in Europe (iHS = − 3.10, $p_{\text{emp}} = 0.002$, 31% of SNPs with $|iHS| > 99$th percentile in a 100 SNP window around the locus, $p_{\text{enrich}} = 0.003$, Fig. 5h), clearly supporting a history of recent adaptation targeting this locus. Overall, while a substantial fraction of population differences may be due to non-genetic factors, our results show that genetic differentiation at miR-QTLs has, in some cases, substantially contributed to population differences in miRNA expression.

### The regulatory impact of miRNA variability on downstream immune responses

Finally, we quantified the extent to which miRNAs contribute to the regulation of immune-related gene expression. To do so, we leveraged mRNA sequencing data obtained for the same individuals, and correlated miRNA expression with mRNA levels of

12,578 genes expressed in our monocyte setting (FPKM > 1) [21], using stability selection (see the "Materials and methods" section). At an 80% probability threshold (~ 1% FDR based on permutations, Additional file 1: Fig. S5a,b), we found that 25–45% of genes were significantly associated with at least one miRNA, with a single gene being independently associated with up to 6 different miRNAs per condition (Fig. 6a and Additional file 6: Table S5). Among conditions, the number of genes associated with a miRNA was slightly higher for viral stimuli (39–44% for R848 and IAV vs. 24–33% for NS, LPS and Pam$_3$CSK$_4$, Fig. 6a). Surprisingly, among the 6009 miRNA-gene associations detected at the basal state, only 43% displayed negative associations, of which 12% presented a known binding site for their associated miRNA (Additional file 6: Table S5). In addition, we found that predicted miRNA targets were depleted in negative correlations with their cognate miRNAs (OR = 0.83, Fisher's $p < 0.02$). When intersecting our miRNA-target predictions with those retrieved from 4 independent databases, we observed no further increase in the strength of correlation or enrichment in negative correlations (Additional file 1: Fig. S5c,d). These observations suggest that the impact

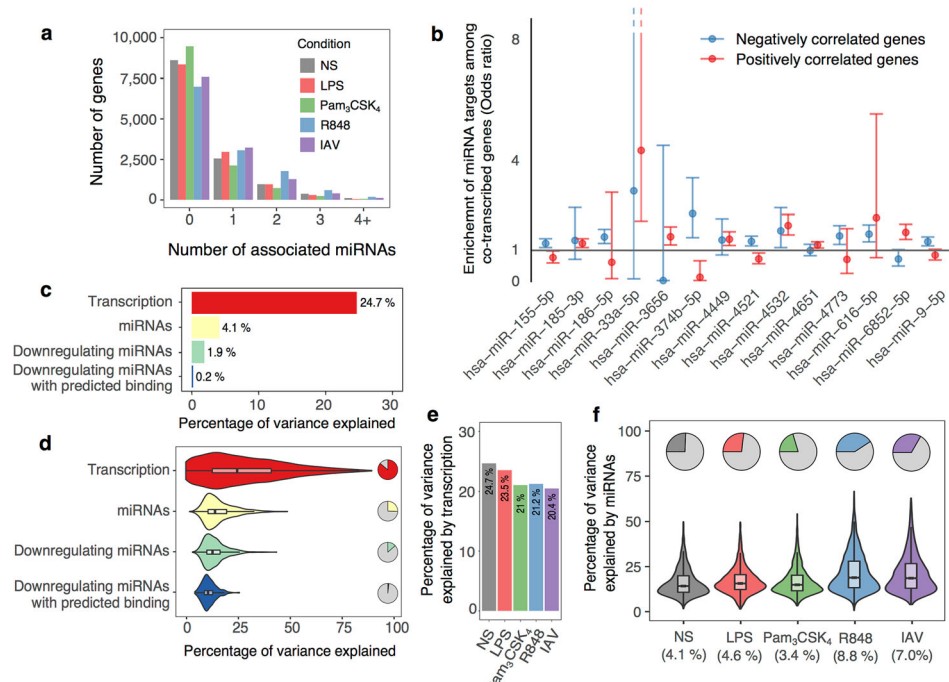

**Fig. 6** Impact of miRNA levels on gene expression. **a** Distribution of the number of associated miRNAs per gene according to the experimental condition. **b** Enrichment or depletion of miRNA targets among genes whose transcription level correlates with miRNA expression (co-transcribed genes). For each miRNA, odds ratios are reported separately for genes whose transcription is positively (red) or negatively (blue) correlated to miRNA expression. Enrichments are displayed only for miRNAs that have a significant enrichment of their targets in either positively or negatively correlated genes (5% FDR). **c, d** Percentage of gene expression variance that is attributable, at the basal state, to either transcription or miRNA variation. For miRNAs, attributable variance is also reported considering only negative associations, or negative associations with predicted binding between the gene and the miRNA. **c** Global percentages (average values across all genes). **d** Distribution at the gene level. Pie charts indicate the percentage of genes associated with transcription or miRNAs. Violin plots display the distribution of the variance attributable to each factor, among significantly-associated genes. **e** Global percentage of variance attributable to transcription according to the experimental condition. **f** Within each experimental condition, pie charts indicate the percentage of genes associated to at least one miRNA. Violin plots display the variance attributable to miRNAs among significantly associated genes

of miRNA-driven transcript degradation on gene expression variability is insufficient to yield a significant enrichment of negative correlations between miRNAs and their targets.

We next hypothesized that the simultaneous interaction of pairs of cooperating miRNAs could be required for effective gene expression regulation. To test this, we focused on a subset of 390 miRNA pairs whose binding sites tend to co-localize < 20 bp away on 3′-UTR regions (see the "Materials and methods" section), and assessed their combined effect on gene expression. Again, the correlation of gene expression with pairs of cooperating miRNAs was found to be independent from the presence or absence of colocalized binding sites in the 3′-UTR region (Additional file 1: Fig. S5e). These results suggest that the reduced inter-individual variability of miRNA expression leads to a minor impact of miRNA-driven transcript degradation on gene expression variability.

Based on these results, we hypothesized that the observed correlations between miRNAs might often be driven by co-transcription rather than miRNA-mediated degradation. To test this hypothesis, we quantified intronic reads that derive from unspliced, nascent transcripts, as a measure of transcription rate [45]. We identified widespread miRNA-gene correlations, each miRNA correlating with transcription rates of ∼ 100 genes at the basal state (min, 1; max, 2680) and up to 173 upon stimulation (min, 1; max, 4137). We found 7 miRNAs that are co-transcribed with their target genes, i.e., present an enrichment of miRNA targets among genes positively correlated in *trans* at the transcription level (Fig. 6b). Among these, the regulator of cholesterol homeostasis miR-33a-5p (OR = 4.3, Fisher's $p < 1.7 \times 10^{-4}$) balances the effect of its host gene, the TF *SREBF2*, on fatty acid synthesis/uptake by repressing the cholesterol transporter ABCA1 [57]. We also identified 7 miRNAs negatively correlated to the transcription of their target genes (Fig. 6b), suggesting a feed-forward loop mechanism, where miRNA downregulation occurs in parallel to the transcription of its target genes to promote rapid expression changes. These include key regulators of the immune response such as the NF-κB inhibitors miR-9-5p (OR = 1.2, Fisher's $p < 5.8 \times 10^{-4}$) and miR-155-5p (OR = 1.3, Fisher's $p < 1.0 \times 10^{-5}$).

Using a variance partitioning approach, we finally quantified the amount of inter-individual variation in gene expression attributed to either gene transcription or miRNA expression [58]. We found that, on average, transcription accounts for 25% of the variance in gene expression at the basal state, with this amount decreasing upon stimulation (min IAV-20%, max LPS-24%, Wilcoxon $p < 2.8 \times 10^{-3}$, Fig. 6c–e). Conversely, miRNA expression variability accounted for only 4.1% of the total variance of expression of their associated genes, and between 3.4% (Pam$_3$CSK$_4$) and 8.8% (R848) upon stimulation (Fig. 6c, d, and f). These figures decreased to ∼ 0.2% when focusing only on negative associations, and disregarding miRNAs with no predicted targets for the gene under consideration (Fig. 6c, d). Testing for the downstream effects on gene expression of the 122 miR-QTLs and 25 isomiR-QTLs, we found no evidence for an enrichment of *trans*-effects compared to random SNPs matched for allele frequency ($\pi_1^{obs} = 0.5\%$, resampling $p = 0.36$). Furthermore, the proportion of *trans*-effects remained negligible when focusing on predicted targets of miRNAs under genetic control ($\pi_1^{obs} = 0.7\%$, 95% confidence Interval (bootstrap) [0.2–1.2%], Additional file 1: Fig. S5f). Altogether, while miRNAs display significant correlations with gene

expression, our results indicate that miRNA variation has a limited impact on gene expression variability.

## Discussion

Several important insights can be drawn from our study. First, we show that upon immune stimulation or infection the miRNA repertoire is subject to important modifications that are not only quantitative, through the modulation of miRNA expression, but also qualitative, through changes in isomiR proportions [9, 35]. Although isomiR modifications can be confounded by cross-mapping artifacts and sequencing errors [28], we reduced here the impact of such technical biases by focusing on frequent, biologically plausible modifications and excluding those that correlate with technical covariates. In doing so, we detected systematic shifts in isomiR proportions occurring primarily upon cellular treatment with viral challenges. While most changes in isomiR usage observed at 6 h of stimulation are of modest effect size, it is possible that they anticipate more drastic modifications occurring later in time, as shown in the case of bacterial infections [9].

Whether induced by stimulation or associated with genetic variants, changes in isomiR usage have the potential to deeply alter miRNA-gene interactions [34, 52]. Although current miRNA prediction algorithms may underestimate the effect of 3′ isomiR modifications on non-canonical binding [34], we find that 19% of isomiRs present significant alterations of their targets relative to the canonical form. Furthermore, for ∼ 7% of miRNAs, modifications alter targets in a non-random manner, i.e., isomiR-specific targets are enriched in specific biological functions. This is illustrated by miR-6503-3p, where viral stimuli induce a shift toward isomiRs with a non-canonical start site that all converge on the targeting of type I-IFN genes, suggesting a role of miR-6503-3p in maintaining a balanced antiviral response. Our work highlights the importance of considering isomiR changes, and not only miRNA expression, when studying the impact of miRNAs on immune responses [34, 35].

Several lines of evidence indicate that strong selective constraints limit miRNA variability in response to immune challenges. First, we observe that both the response of miRNAs to immune activation and its variability between populations is much more nuanced than that of protein-coding genes. Second, we report a limited number of miRNAs, as well as isomiRs, whose expression levels are associated with miR-QTLs and isomiR-QTLs, even among highly expressed miRNAs. Finally, in contrast with protein-coding eQTLs, which are largely context-dependent and thus variable across stimuli, we find that the detected miR-QTLs are mostly unaffected by immune stimulation. Our analyses thus support the notion that the miRNA system is little tolerant to genetic variation modulating its response to stimulation.

Despite the global constraints driving miRNA variability, we find marked differences in miRNA expression between individuals of African and European ancestry, with ∼ 25% of miRNAs per condition being differentially expressed between populations. The limited genetic control of miRNA expression, with respect to protein-coding genes, indicates that most of the observed population differences are attributable either to *trans*-acting genetic factors regulating the miRNA biogenesis/decay pathways or to non-genetic factors. While over 60% of expression differences related to ancestry are unaffected by stimulation, we identified 9 miRNAs that present population differences uniquely upon immune challenge, all presenting a trend toward stronger inducibility in

Africans. Of these, 5 have been previously reported to correlate with induction of LPS tolerance in mice [59], including key immune regulators such as miR-155-5p and miR-222-3p. This suggests that population differences in miRNA responses may lead to a weaker innate immunity response to secondary challenges among African-descent individuals.

Despite *cis*-genetic factors account for only a fraction of ancestry-related differences in miRNA expression, these explain ~ 60% of the observed population differences for miRNAs associated with a miR-QTL. For example, the frequency of the disease-associated variant rs290164 is sufficiently different between populations ($\Delta DAF = 28\%$) to explain ~ 76% of the differences in isomiR ratios for miR-146a-3p. Furthermore, we identify a European-specific variant (rs12881760), which controls a cluster of 12 miR-NAs in *cis*, that displays extreme population differentiation ($F_{ST}$) with Africans (top 0.2% of $F_{ST}$) and East Asians (top 0.004% of $F_{ST}$). The adaptive nature of this variant in Europeans is supported by a strong enrichment of |iHS| outliers at this locus that, moreover, has been associated with platelet parameters, likely underlying differences in platelet activity between European- and African-ancestry individuals [60]. Interestingly, an independent event of positive selection targeting the same miRNA-rich cluster has been detected in Asian populations [61], highlighting the adaptive role of this locus in populations of non-African ancestry.

Finally, our study-design allowed us to assess the relative contribution of transcriptional regulation and miRNA-mediated degradation on downstream immune responses. While we find a strong effect of transcription rate on gene expression, our model predicts that miRNA-mediated degradation accounts for < 0.2% of the variation in gene expression. This result, together with the lack of measurable effects of miR-QTLs on gene expression, suggests that individual miRNAs have only a limited impact on population differences of mRNA expression levels. This could be explained both by small effects of individual miRNAs on gene expression and by a reduced variability of miRNA expression itself, as suggested by their limited genetic control. Yet, this does not preclude an important role of miRNAs in the regulation of gene expression through the aggregate contribution of a large number of miRNAs, or in the fine-tuning of protein responses through translational inhibition.

Furthermore, our results are consistent with previous reports of low levels of miRNA-mRNA correlations [8, 13, 16, 62] and provide a model to explain the frequent occurrence of positive correlations between miRNA expression and that of their predicted targets. Indeed, we highlight several cases where miRNA expression is correlated with the transcription of their targets, creating either feedback loops, as for miR-33a-5p, or feed-forward loops, as for the TLR-induced miR-9-5p and miR-155-5p. When adjusting for transcription rate, miRNA expression captures 3 to 6% of the variation in gene expression on average. This could reflect either an indirect contribution of miR-NAs to immune response variability through translation inhibition of key immune regulators or residual co-transcription due to the dynamic nature of gene transcription.

## Conclusion

Together, this study shows that genetic and non-genetic factors contribute to marked population differences in miRNA abundance and isomiR ratios. Yet, we also show that such differences have a moderate impact on the transcriptional landscape of immune

cells at 6 h, suggesting that the consequences of miRNA deregulation may be most visible at later stages of the immune response. Overall, our study reports a large set of miRNAs and isomiRs that present differential responses across bacterial- and viral-like challenges and/or between populations of different ancestry, during an early time window of the innate immune response. In doing so, it constitutes a useful resource for evaluating their role in shaping variation of immune response to infection and disease susceptibility both at the individual and population levels.

## Materials and methods

### Samples and dataset

Biological samples were generated as part of the EvoImmunoPop project [21]. Briefly, the EvoImmunoPop cohort is composed of 200 healthy, male participants of self-reported African and European descent, recruited in Belgium (100 individuals of each population). For all individuals, total RNAs from CD14-positive cells were treated for 6 h with five conditions of stimulation (resting, LPS, $Pam_3CSK_4$, R848, and IAV A/USSR/90/1977). Genotyping was performed using both Illumina HumanOmni5-Quad BeadChips and whole-exome sequencing with the Nextera Rapid Capture Expanded Exome kit. Stringent quality control and imputation procedures were applied [21], leading to a final set of 19,619,457 SNPs, of which 9,854,620 SNPs had a minor allele frequency (MAF) greater than 5% in either population of our cohort. Regarding the mRNA sequencing dataset, libraries from total RNA samples and transcriptome sequencing were performed using TruSeq RNA Sample Prep Kit v2 for mRNA library construction, TruSeq SR Cluster Kit v3-HS for cluster generation, and TruSeq SBS kit v3-HS for sequencing on an Illumina HiSeq2000 platform. In total, an average of 34.4 million 101-bp single-end reads per sample (min: 27.7-max: 94.8 million reads) were obtained [21]. High-density genotyping and exome sequencing data, and the mRNA sequencing data, used in this study are available in the European Genome-Phenome Archive (EGAS00001001895).

### Small-RNA library preparation and sequencing

Total RNA samples have been used to generate miRNA sequencing data. Low molecular weight RNA fragments were selected by gel excision (targeting fragments of ~ 22 bp), and sequencing libraries were prepared using the Illumina TruSeq small RNA library prep Kit. Indexed cDNA libraries were then pooled by groups of 18 (in equimolar amounts) and sequenced with single-end 50 bp reads on the Illumina HiSeq2000. After exclusion of one sample that yielded less than 1.8 million read counts, we obtained an average of 12.4 million raw reads per sample with a minimum yield of 8.0 million reads.

### Pre-processing of raw sequencing reads

Sequences matching the 3′ adaptor sequence were identified and trimmed, using fastx_clipper version 0.0.13 with the following options –l 0 –n –M 10, to require a minimum adapter alignment length of 10 base pairs, while keeping all sequences regardless of their length or presence of unknown nucleotides. This led to the exclusion of ~ 2% of reads per sample. Final read lengths ranged from 1 to 42 bases. We confirmed that all samples had average base quality (Q) values > 30 at all positions, and that per-base GC distributions were within expected ranges. We further checked that read length distributions showed

an enrichment of ~ 22 bases-long reads for all samples, consistent with expectations for mammalian miRNAs (~ 22 bases), and discarded reads shorter than 18 or longer than 26 bases. After these filtering steps, we obtained an average of 8.8 million (minimum 4.1 million) short reads per sample, which were used for small RNA quantification.

### Sequence alignment

Sequences were aligned to the human reference genome (build GRCh37/hg19) using bowtie (version 1.1.1) [63]. We mapped reads allowing for 2 mismatches (–v 2) and reported all best alignments for reads that mapped equally well to more than one genomic location (-a--best--strata). We suppressed reads with more than 50 possible alignments (-m 50). On average, ~ 97% of reads aligned to the genome (min 90%), of which 59% overlapped a known miRNA. Due to their reduced size, miRNAs are known to be susceptible to cross-mapping, i.e., spurious read alignments to other related miRNAs with strong sequence similarity [28]. In the present dataset, around 65% of reads aligning to known miRNAs had more than one possible alignment on the genome. To mitigate the impact of such cross-mapping on miRNA quantification, we used a correction strategy that assigns weights to each of the candidate mapping loci of multiply aligning reads, based on local expression levels and mismatches in the alignment [28], allowing to distinguish true miRNA reads from likely alignment errors.

### Quantification of miRNA expression

We extracted reads aligning to annotated mature miRNA sequences (miRBase v20) [64] with at least 75% overlap using BEDTools [65] and divided counts per million associated of each miRNA by the total number of miRNA mapping reads to obtain comparable numbers across all libraries. In addition, we used DESeq2 (version 1.20) [66] to compute size factors associated to each library and normalize miRNA counts per million across libraries. Size factor normalization prevents systematic shifts in log counts between samples (and thus conditions) by subtracting, from each log-transformed sample, the median across genes of the difference between the log counts of the sample and the average log counts across all samples. We then removed lowly, or sporadically, expressed miRNAs by keeping only those with counts of greater than 1 read per million on average across all experimental conditions, leading to a final set of 658 miRNAs across 736 loci. We then added a pseudo-count of 1 RPM to all miRNAs, and $\log_2$ transformed the data to stabilize the variance of miRNA expression. Linear models were then used to adjust $\log_2$ transformed counts for technical confounders, such as mean read length of the library (after clipping), or mean GC content of miRNA-aligned reads. Batch effect induced by date of experiment and library preparation were sequentially removed using ComBat [67].

### Assessment of isomiR diversity

For the analyses at the isomiR level, reads aligning to annotated mature miRNA sequences were extracted as described above, and each unique sequence with a mean expression of > 1 count per sample was treated as a separate isomiR. miRNA sequences presenting less than 1 count per sample on average were discarded, and read counts were normalized using the same approach applied for total miRNA expression. We also

removed reads where at least one nucleotide could not be called. For each miRNA, the canonical sequence was defined according to miRBase v.20 [64] and similarity with canonical sequence at nucleotides 2–7 was used to distinguish *canonical seed* isomiRs from *non-canonical seed* isomiRs. We next classified isoform modifications into three main categories, each subdivided into subtypes of miRNA modifications: (i) *changes in start site*, subdivided in *5′ extension* and *5′ reduction*; (ii) *template changes in end site*, subdivided in *3′ extension* and *3′ reduction*; and (iii) *non-template 3′ additions*, subdivided into *3′ adenylation* and *3'uridylation*. Finally, we quantified, for each miRNA, the frequency of each type of modification and used these quantities for all downstream analyses. These frequencies were then averaged across miRNAs, to provide global estimates of the frequency of miRNA modification events across samples. After the initial quantification of isomiR diversity, isomiRs that differed only by an internal substitution or a non-canonical terminal addition were merged for downstream analysis, to reduce the effect of sequencing errors.

### Quantification of gene expression levels and transcription rate

RNA-seq reads were aligned to hg19 using Tophat2 [68] and gene expression values (FPKM) were computed with CuffDiff [68] based on Ensembl v70. Samples with uneven gene coverage were excluded leaving a total of 969 samples with both miRNAs and protein-coding gene expression. Gene expression values were log-transformed (with an offset of 1) and corrected for GC content and 5′/3′coverage bias, as well as experiment and library preparation date using linear models and ComBat [69]. Only the 12,578 genes with mean FPKM > 1 were kept for downstream analyses. Further details on gene expression quantification, QC, and normalization can be found elsewhere [21]. Transcription rates were estimated based on the number of nascent unspliced transcripts [45]. Namely, for each gene, we used HT-Seq [70] to compute the average number of reads mapping to the gene after exclusion of all exonic regions. This number of intronic reads was then divided by the total length of introns, to yield a mean intronic coverage that was used as a proxy of the transcription rate. For each gene, inverse-normal rank transformation was applied to gene expression levels and transcription rate to reduce the impact of outlier values in downstream analyses.

### Differential expression and isomiR analysis

To identify miRNAs that are differentially expressed upon stimulation, we transformed miRNAs counts using an inverse normal rank-transformation and fitted a linear mixed model of the form $\{y_{ij} = a_i + b.\mathbb{I}_j^{stim} + \varepsilon_{ij}\}$, where $y_{ij}$ is the transformed counts of individual $i$ in condition $j$, $a_i$ is a random effect capturing the inter-individual variability in miRNA expression, $b$ is the effect of stimulation on miRNA expression, $\mathbb{I}_j^{stim}$ is an indicator variable equal to 0 for the non-stimulated samples and 1 for stimulated samples, and $\varepsilon_{ij}$ are the residuals. Significance was assessed by maximum likelihood test and a global Benjamini and Hochberg FDR-correction across all 4 stimuli. Only changes in miRNA expression with corrected $p$-value < 0.01 were considered significant. To detect a significant change in isomiR ratios, we employed a similar approach using isomiRs ratios instead of miRNA read counts.

## Sharing of effects across conditions

To assess the similarity of miRNA response across stimuli, we focused on all miRNAs and isomiRs that were detected to respond to stimulation in at least one condition. We then used a likelihood-based model selection framework [49], assuming that miRNAs respond to only a subset of stimuli, and identified the most likely subset of stimuli by jointly modeling rank-transformed miRNA expression, or isomiR ratios, across all 5 conditions. Specifically, for each stimulus $j$, we assigned an indicator variable $\gamma_j$ equal to 1 if a given miRNA responds to the stimulus and 0 otherwise. Then, for each of the 15 non-null combinations of stimuli $(\gamma_j)_{j \in \{1, 2, 3, 4\}}$, we fitted a linear mixed model, as previously performed in each condition { $y_{ij} = a_i + b. \gamma_j + \varepsilon_{ij}$ }, with $a_i$ a random effect capturing the inter-individual variability in miRNA expression or isomiR ratio, $b$ is the effect of stimulation and $\varepsilon_{ij}$ the residuals, and assigned a probability to each model $m$ as

$$Prob(\text{Model } m) = \frac{Likelihood_m}{\sum_{k=1}^{15} Likelihood_k}$$

## Detection of miRNA-QTLs and isomiR-QTLs

To identify genetic variants associated with miRNA expression or isomiR ratios, i.e., miR-QTLs and isomiR-QTLs, we focused on the 598 miRNAs that could be uniquely assigned to a single genomic locus and considered a set of 9,854,620 genetic variants with a minor allele frequency (MAF) > 0.05 in either African- or European-ancestry groups, of which 1,981,401 were located < 1 Mb from one the 598 mature miRNAs. We used *MatrixEQTL* [71] to map miR-QTLs within a 1 Mb window on each side of mature miRNAs. miR-QTL mapping was performed separately for each condition, merging both populations and including an indicator variable to control for the effect of population on miRNA expression. miRNA counts per million values and isomiR ratios were rank-transformed to a normal distribution before mapping, to reduce the impact of outliers. FDR was computed by mapping miR/isomiR-QTLs on 100 permuted datasets, in which genotypes were randomly permuted within each population. We then kept, for each permutated dataset, the most significant $p$ value per miRNA or isomiR, across all conditions, and computed the FDR associated with various $p$ value thresholds ranging from $10^{-3}$ to $10^{-50}$. We subsequently selected the $p$-value threshold that provided a 5% FDR ($p < 10^{-6}$).

## Comparison of miR-QTLs to protein-coding gene eQTLs

To compare the degree of genetic control of miRNA expression to that of protein-coding genes and long non-coding RNAs, we used the *MatrixEQTL* package [71] to perform eQTL analyses, as performed for miR-QTLs. We considered a 1 Mb window around the TSS of each gene and tested all frequent SNPs (i.e., with MAF > 5% in either population) for association with gene expression. eQTLs were tested in both populations combined on rank transformed gene expression values, adjusting for population. FDR was computed based on 100 permutations as described for miRNAs. When comparing frequency of eQTLs and miR-QTLs, a joint FDR was re-computed considering

miRNAs, protein-coding genes, and long noncoding RNAs together, to ensure similar power for the detection of miR-QTLs and eQTLs. Protein-coding genes were then assigned to various categories based on Exac pLI scores (LOF intolerant – pLI > 0. 9, recessive pRec > 0.9, neutral pNull > 0.9) [72] or GO annotations (TF – GO:0003700). In addition, to ensure that differences in read coverage between miRNAs and protein-coding genes were not responsible for a lower power to detect miR-QTLs/eQTLs, we evaluated the impact of read coverage on the detection of miR-QTLs and eQTLs (Additional file 1: Fig. S4a,b). We removed from our analyses miRNAs and genes with < 50 supporting reads on average.

### Sharing of miR-QTLs and eQTLs across conditions

When comparing miR-QTLs across conditions, we used a likelihood-based model selection framework to increase power for detection of shared effects. Namely, for each SNP-miRNA pair, rank-transformed miRNA expression, or isomiR ratios, $y_{ij}$ were modeled jointly across all 5 conditions. An indicator variable $\gamma_j$ was defined as 1 if the miRNA is under genetic control in condition j and 0 otherwise. Then, for each of the 31 non-null combinations of stimuli $(\gamma_j)_{j \in \{1, 2, 3, 4, 5\}}$, we fitted a linear model of the form { $y_{ij} = a_{jp} + b. \ \gamma_j SNP_i + \varepsilon_{ij}$ }, with $a_{jp}$ the mean expression of the miRNA or isomiR in condition j and population p, $SNP_i$ the number of minor alleles carried by individual i, b the mean effect of the SNP in conditions where it is active, and $\varepsilon_{ij}$ the residuals. Each model was then assigned a probability as follows:

$$Prob(\text{Model } m) = \frac{Likelihood_m}{\sum\limits_{k=1}^{31} Likelihood_k}$$, and the model with the highest probability was

retained. The same approach was applied to assess the sharing of eQTLs across conditions, using rank transformed mRNA levels (FPKM) of protein-coding genes instead of miRNA expression.

To assess whether the higher stability of miRNAs upon stimulation, relative to protein-coding genes, could contribute to the lower occurrence of condition-dependent miR-QTLs, we considered all eQTLs and miR-QTLs and used logistic regression to model the probability that their effect is condition-dependent (i.e., observed in only a subset of experimental conditions) as a function of the nature of the molecular trait (protein-coding gene or miRNA) and its maximal absolute fold change in response to stimulation.

In addition, to identify response-miR-QTLs, we tested for significant differences in effect size of miR-QTLs between the stimulated and non-stimulated state, using an interaction test. Rank-transformed miRNA expression, or isomiR ratios, $y_{ij}$ are decomposed between $a_{jp}$ the mean expression of the miRNA or isomiR in condition j and population p, the effect of the SNP at basal state b, and the differences in effect size between basal and stimulated state c.

$$y_{ij} = a_{jp} + b.SNP_i + c.SNP_i.\mathbb{I}_j^{stim} + \varepsilon_{ij}$$

The significance of the interaction was then tested by a Student t test for H$_0$: {c = 0}.

### Annotation of miRNA TSS and miR-QTLs

Transcription start sites (TSS) of miRNAs were obtained from Fantom5 data, based on [73], together with their conservation levels (mean PhastCons of promoter region).

Hairpin coordinates were retrieved from mirBase V20 [64]. MiR-QTLs for which TSS information was available were then classified based on their location relative to the TSS and hairpin. Namely, miR-QTLs were first classified as *miRNA-altering* or *hairpin-altering* if they overlapped the sequence of the mature miRNA or its associated hairpin. Then, we computed, for each miR-QTL, the distance between the SNP and both the hairpin and the TSS of the associated pri-miRNA. MiR-QTLs that were located less than 20 kb from the TSS or the hairpin were annotated as *hairpin-* or *TSS-flanking*, according to the feature from which they were the closest. Finally, miR-QTLs located > 20 kb from both TSS and hairpin were annotated as *Distant*. Overlap of miR-QTLs (and eQTLs) with promoters and enhancers was assessed based on Epigenomic Roadmap data, using the ChromHMM segmentation of tissue E029 (CD14$^+$ monocytes) [74]. Specifically, regions assigned to class 1 and 2 (Active TSS and Active TSS Flank) were considered as promoters, and class 6 and 7 (Enhancers and Genic enhancers) were considered as enhancers. MiR-QTL and eQTL peak SNPs were then compared to the set of all frequent SNPs (MAF > 5%) considered in our study.

Intronic miRNAs were defined based on their overlap with the 12,578 genes that are expressed at FPKM > 1 in any of the 5 experimental conditions [21]. To assess the impact of host-gene eQTLs on intronic miRNAs, we considered the condition where the host gene eQTL is the most significant and correlated the peak SNP of the eQTL with miRNA expression in the same condition. The 81 *p*-values that we obtained were then adjusted for multiple testing with Benjamini-Hochberg correction, using a 1% FDR threshold. Overlap of miR-QTLs of intronic miRNAs with eQTLs of their host genes was assessed by computing the linkage disequilibrium (LD), as measured by $r^2$, between genotypes of the peak miR-QTL and eQTL SNPs, across European and African individuals combined. Finally, for the 6 miR-QTLs in high LD with an eQTL ($r^2 > 0.8$), we adapted a likelihood-based causality model selection approach previously developed [55]. This approach was used to infer the most likely model of causality between a model where (i) the genotype impacts miRNA expression through its effect on expression of the host gene, (ii) the genotype impacts host gene and miRNA expression independently, and (iii) the genotype impacts host gene expression through its effect on the miRNA (feedback loop). Specifically, each model was assigned likelihood as follows:

$$L_1 = Lik(miRNA|host\ gene).Lik(host\ gene|genotype)$$

$$L_2 = Lik(miRNA|genotype).Lik(host\ gene|genotype)$$

$$L_3 = Lik(host\ gene|miRNA).Lik(miRNA|genotype)$$

Where $Lik(Y|X)$ is the likelihood of the linear model of the form:

$$Y = a + b.X + c.population + \varepsilon$$

The model with the highest likelihood was then considered as the most likely causality model, accounting for the colocalization of the miR-QTL and host gene eQTL.

### Population differences in miRNA and isomiR expression

To identify miRNAs that are differentially expressed between populations, we applied Student's *t* test to inverse normal rank-transformed miRNAs counts within each condition separately, comparing African- to European-ancestry individuals. A global

Benjamini and Hochberg FDR correction was applied across all 5 conditions to evaluate significance. Only changes in miRNA expression with corrected $p$-value $< 0.01$ were considered as significant. A similar approach was used to test for population differences in isomiR levels, using isomiRs ratios, instead of miRNA read counts. Sharing of population differences among conditions was assessed using a model selection framework similar to the one used to assess sharing of mir-QTLs. For each individual $i$ and condition $j$, we assigned an indicator variable $\gamma_j$ equal to 1 if a miRNA is differentially expressed between populations in that condition and 0 otherwise. Then, for each of the 31 non-null combinations of conditions $(\gamma_j)_{j \in \{1, 2, 3, 4, 5\}}$, we fitted a linear model $\{y_{ij} = a_j + b.\gamma_j.\mathbb{I}_i^{pop} + \varepsilon_{ij}\}$, with $a_j$ the mean expression of the miRNA in condition $j$ across African-ancestry individuals, $b$ the mean difference in miRNA expression between European- and African-ancestry individuals, and $\varepsilon_{ij}$ a normally distributed residual. Each possible model was then assigned a probability $m$ as

$$Prob(\text{Model } m) = \frac{Likelihood_m}{\sum_{k=1}^{31} Likelihood_k} \text{ and the most likely model was retained.}$$

### Assignment of miRNA/isomiR targets

miRNA targets were predicted using miRanda v3.3a [51], providing canonical sequences obtained from miRBase V20 [64] as input and 3′ UTR sequence of known transcripts based on Ensembl V70. Defaults settings were used for target prediction, and a gene was considered as targeted by a miRNA if at least one of its annotated transcripts had a predicted binding site for the miRNA. Prediction of isomiR targets was performed in a similar manner, using the isomiR sequence instead of the canonical sequence. IsomiR targets were compared to targets of the canonical isomiR. For each isomiR with more than 30 targets lost or gained, compared to the canonical isomiR, enrichments of targets in specific biological functions were assessed using goseq [75], adjusting for the length of the 3′-UTR regions.

To assess the robustness of miRNA targets predictions, we overlapped targets predicted by miRanda with target predictions obtained with 4 alternative methods, available in public databases. Specifically, we used (i) conserved and non-conserved targets from *targetScan v7.1* [76], which predicts binding based on the presence of matches to the miRNA seed, seed type, and local nucleotide context; (ii) targets from *MIRZA* [42], which predicts binding based on biophysical model allowing to capture non-canonical binding; (iii) targets from *MIRZA-G* [41], which integrates *MIRZA* targets with information on local nucleotide content, position on the 3′UTR, mRNA structure, and target site conservation to predict regulatory potential; and (iv) targets from *mirTarget* [43], which uses machine learning based on miRNA over-expression experiments to predict miRNA targets with regulatory potential. For each miRNA-gene interaction, we counted the number of algorithms that predicted the interaction and used this number as a measure of confidence in the miRNA-gene interaction.

### Assessment of miRNA-mRNA correlations

To identify likely miRNA-gene interactions occurring in each condition, we modeled gene expression as a function of miRNA levels, using population as a covariate. All miRNAs

were introduced simultaneously in the model and an elastic net penalty [77] was set on the miRNA effects to make the model identifiable, leading to the following model

$$Expr = a + b.population + \sum_{j=1}^{n} c_j.miRNA_j + \varepsilon$$

with

$$\left( \sum_{j=1}^{n} |c_j| + \sum_{j=1}^{n} c_j^2 \right) < \lambda$$

Here, *Expr* is the vector of gene expression across all samples from the condition under study, *population* is an indicator variable representing the population of origin, $(miRNA_j)_{j=1..n}$ are the vectors of expression of the 658 expressed miRNAs, and $\varepsilon$ is a random Gaussian noise. *a* denotes the mean expression in the reference population, *b* and $(c_j)_{j=1..n}$, are parameters capturing the effect of population, and miRNAs on gene expression. $\lambda$ is a constant value that captures the amount on constraint on the effect of miRNAs included in the model.

Using this model, we performed stability selection [78] to select miRNAs that have a significant effect on gene expression with high probability. Briefly, stability selection consists in performing repeated sub-samplings of the data, typically considering only half of the initial data, and selecting the first Q miRNAs with non-null $c_j$ coefficients across increasing values of $\lambda$. Under the reasoning that only miRNAs with a true effect on gene expression will be consistently selected across subsamplings, we can then use the frequency at which a miRNA is kept in the model as the posterior probability that this miRNA has a significant impact on gene expression. We performed 100 resamplings with varying values of Q (from 3 to 60, by steps of 3) and estimated, for each value of Q, the probability of each miRNA to be included in the model. Then, for each value of Q, we randomly permuted the data and repeated the procedure to obtain the distribution of inclusion probabilities, under a model where gene expression is independent from miRNA levels. A single permutation of the data was performed for each Q, and the null distribution was estimated across all 12,578 genes × 658 miRNAs. Based on this null distribution, we computed the FDR associated with a given probability threshold, as the ratio between the number of significant gene-miRNA pairs that exceed that probability threshold in the permuted and non-permuted data set. We found that setting Q = 9 maximized the number of significant associations at an FDR of either 1, 5, or 10% and used this value for all subsequent analyses. We then considered as significant all miRNAs that reach a posterior probability of 0.8, which is equivalent to an FDR of ~ 1.3% based on our permutation setting (Additional file 1: Fig. S5a,b).

### miRNA-miRNA interactions and their impact on gene expression

To establish the impact of miRNA interactions on the regulation of gene expression, we first searched for pairs of miRNAs that have binding sites < 20 bp apart of each other on the same mRNAs, more often than expected by chance. Specifically, we counted for each pair of miRNAs the number of genes with binding sites from the two

miRNAs located < 20 bp away on at least one of their transcripts, and tested whether this number exceeded the expected number of shared binding sites, based on a Poisson distribution with parameter $\lambda$ given by

$$\lambda = N_{\text{gene}} . P_{\text{miR}_1}^{\text{target}} . P_{\text{miR}_2}^{\text{target}} . P_{\text{coloc}}$$

where $N_{\text{gene}}$ is the total number of genes, $P_{\text{miR}_i}^{\text{target}}$ is the probability that a random gene is targeted by $\text{miR}_i$—estimated as the ratio between the number of target of $\text{miR}_i$ and the total number of gene tested—and $P_{\text{coloc}}$ is the average probability that the targets of both miRNAs co-localize—i.e., are located < 20 bp away—given that they both target the selected gene. $P_{\text{coloc}}$ is approximated by the ratio $P_{\text{coloc}} \approx \frac{20}{\text{average}(3' \text{UTR length})}$.

We then applied a Benjamini-Hochberg correction to the resulting *p*-values and selected for follow-up the set of 390 miRNAs pairs that (i) passed a 1% FDR threshold and (ii) had more than 100 expressed genes with co-localized targets. To assess the impact of these interacting miRNAs, we reasoned that under a model where both miRNAs need to be present for the degradation of the mRNA, mRNA expression will depend either on the expression of the miRNA with the lowest expression (if the other miRNA is in large excess) or on the product of the amount of both miRNAs if they are expressed at similar levels. For each condition, we thus assessed the combined impact of each pair of miRNA of their common targets, by considering the following linear model and testing for the global null hypothesis $H_0$: $c_1 = c_2 = c_{12} = 0$.

$$Expr = a + b.population + c_1 miRNA_1 + c_2 miRNA_2 + c_{12} miRNA_1.miRNA_2 + \varepsilon$$

in this model, *Expr* is the expression of the target gene, *population* is a binary variable indicating the population to which the individual belongs, $miRNA_1$ and $miRNA_2$ refer to the expression of the interacting miRNAs, and $\varepsilon$ are normally distributed residuals. We then compared the proportion of genes for which $H_0$ is false (as estimated by *pi0est*(.) function from the *qvalue* package) between genes that are predicted as targets of both miRNAs, and the remainder of the genome, based on a set of 100 bootstrap replicates.

### Correlation between miRNAs and transcription

Correlation between miRNAs and transcription rate was obtained using the *Matrix-EQTL* package [71], providing miRNAs instead of genotypes and adjusting for population. All associations where the miRNA was located less than 1 kb away from the gene were discarded, and a 5% FDR was used to declare associations as significant. For each miRNA, significantly associated genes were split into *negatively* and *positively* correlated genes according to the sign of the corresponding β parameter. We then tested each set of associated genes for enrichment in predicted binding sites obtained from miRanda [51] compared to the set of all transcribed genes with at least one predicted miRNA binding site. Benjamini-Hochberg correction was applied across all miRNAs for both positive and negative correlations, and only enrichments passing a 5% FDR were retained.

### Trans-effects of miRNA-QTLs on gene expression

To assess the effect of miR-QTLs and isomiR-QTLs on gene expression, we considered the set of 118 unique SNPs with an effect on miRNA expression or isomiRs ratios and tested for *trans*-associations with genes located > 1 Mb away from these SNPs. We then used the *pi0est(.)* function from the *qvalue* package to estimate the percentage $\pi_1^{obs}$ of genes associated with a miR-QTL or isomiR-QTL across all 5 conditions, based on the shape of the *p*-value distribution. Finally, we repeated the same analysis for 100 random samples of SNPs, matched for minor allele frequency (using MAF bins of 5%) and computed a resampling *p*-value by counting the frequency at which the percentage $\pi_1^{obs}$ of genes associated to miR-QTLs and isomiR-QTLs exceeded the $\pi_1$ value estimated from sets of randomly selected SNPs. Furthermore, using the same approach, we measured the proportion of genes associated with a miR-QTL or isomiR-QTL among genes that are predicted to be targets of the miRNAs whose expression is controlled by the locus, for various levels of confidence in the miRNA-gene interaction.

### Relative contribution of transcription and miRNAs to gene expression variability

To account for co-transcription when assessing miRNA-gene correlations, we repeated our stability selection approach adding transcription as a covariate in the model. The final model can thus be written as

$$Expr = a_p + b.transcription + \sum_{j=1}^{n} c_j.miRNA_j + \varepsilon$$

with

$$\left( |b| + \sum_{j=1}^{n} |c_j| + b^2 + \sum_{j=1}^{n} c_j^2 \right) < \lambda$$

Here, *Expr* and *transcription* are the vectors of gene expression and transcription rate across all samples from the condition under study, $(miRNA_j)_{j=1..n}$ are the vectors of expression of the 658 expressed miRNAs, and $\varepsilon$ is a random Gaussian noise. $a_p$ denotes the mean expression in population $p$ and $b$ and $(c_{j,})_{j=1..n}$ are parameters capturing the effect of transcription and miRNAs on gene expression. $\lambda$ is a constant value that captures the amount on constraint on miRNAs that are included in the model.

After identifying miRNAs that have a significant effect on gene expression, miRNA effect sizes were assessed using CAR scores as implemented in the *care* package [58]. Briefly, CAR scores (noted $\omega$) are a variation of partial correlations that allows to measure correlations between one or more covariates and a response variable, while adjusting each covariate for the effect all other covariates. More importantly, the squared CAR scores ($\omega^2$) sum to the total percentage of variance explained by the model ($R^2$), allowing to interpret the square of each individual CAR score as the percentage of variance explained by the associated covariate, when adjusting for all other covariates. To evaluate the variance explained by a subset of miRNAs (i.e., negatively correlated miRNAs or negatively correlated miRNAs with a known binding site to the gene), we considered the sum of $\omega^2$ over all miRNAs of that subset (using the sign of $\omega$, to identify negative correlations).

## Supplementary information

---

**Additional file 1: Fig. S1**. Quality and pre-processing of small RNA sequencing data. **Fig. S2**. IsomiR diversity and rate of miRNA modifications. **Fig. S3**. Sources of isomiR variation upon immune activation. **Fig. S4**. Genetic basis of miRNA expression. **Fig. S5**. Detection of miRNA-mRNA correlations.

**Additional file 2: Table S1**. List of miRNAs, isomiRs, and miRNA modifications.

**Additional file 3: Table S2**. Impact of immune stimulation of miRNA and isomiR expression

**Additional file 4: able S3**. Genetic control of miRNA and isomiR expression.

**Additional file 5: Table S4**. Population differences in miRNA and isomiR expression.

**Additional file 6: Table S5**. List of miRNA-correlated genes across the 5 experimental conditions.

**Additional file 7.** Review history.

---

### Acknowledgements
We thank Macrogen Inc. for the use of their RNA-sequencing facilities.

### Peer review information

### Review history
The review history is available as Additional file 7.

### Authors' contributions
M.R. and K.J.S conceived the analysis pipeline; M.R. supervised the analyses; M.R., M.S., and K.J.S. analyzed and interpreted the data; J.P. and H.Q. designed and performed the experiments; L.Q.M. interpreted the data, conceived and supervised the study, and obtained funding; and M.R. and L.Q.M. wrote the manuscript, with inputs from all authors. All authors read and approved the final manuscript.

### Funding
The laboratory of L.Q.-M. is supported by the Institut Pasteur, the Collège de France, the French Government's *Investissement d'Avenir* program, *Laboratoires d'Excellence* "Integrative Biology of Emerging Infectious Diseases" (ANR-10-LABX-62-IBEID) and "Milieu Intérieur" (ANR-10-LABX-69-01), and the *Fondation pour la Recherche Médicale* (Equipe FRM DEQ20180339214). This project was funded by the European Research Council under the European Union's Seventh Framework Programme (FP/2007–2013)/ERC grant agreement 281297.

### Availability of data and materials
The miRNA-sequencing data generated in this study have been deposited in the European Genome-phenome Archive (EGA) under accession code EGAS00001004192 [79].
Genotyping, Exome sequencing and mRNA-sequencing data used in this study are available in the European Genome-phenome Archive (EGA) under accession code EGAS00001001895 [80]. All scripts used for this study have been deposited on Github: https://github.com/mrotival/EvoImmunoPop_miRNAs [81].

### Ethics approval and consent to participate
Human primary monocytes were obtained from healthy volunteers who gave informed consent. This study was approved by the Ethics Board of Institut Pasteur (EVOIMMUNOPOP-281297) and the relevant French authorities (CPP, CCITRS, and CNIL). All experimental methods were conducted in accordance with the Declaration of Helsinki principles.

### Consent for publication
Not applicable.

### Competing interests
The authors declare that they have no competing interests.

### Author details
[1]Human Evolutionary Genetics Unit, Institut Pasteur, CNRS UMR 2000, 75015 Paris, France. [2]Broad Institute of MIT and Harvard, Cambridge, MA, USA. [3]Department of Organismic and Evolutionary Biology, Harvard University, Cambridge, MA 02138, USA. [4]Sorbonne Universités, École Doctorale Complexité du Vivant, 75005 Paris, France. [5]Present Address: DIACCURATE, Institut Pasteur, 75015 Paris, France. [6]Present Address: UMR7206, Muséum National d'Histoire Naturelle, CNRS, Université Paris Diderot, 75016 Paris, France. [7]Chair Human Genomics and Evolution, Collège de France, 75005 Paris, France.

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

## 

