## [**Additional file 7.** Review history. · Genome Biology]

Review History

First round of review

Reviewer 1

Are you able to assess all statistics in the manuscript, including the appropriateness of statistical tests used?

No, I do not feel adequately qualified to assess the statistics.

Comments to author:

The manuscript is well written and comprehensively details the cataloging of miRNA sequence variants across an impressively large number of samples (monocytes from 200 individuals, treated separately with 4 stimuli). The area of isomiRs is potentially significant as miRNA sequence variation is extensive, though there remains a lack of understanding how widely regulated these differences are and what their functional consequences are likely to be. I have several comments for the authors to consider, though my main critique is the impact of the cataloging of such variants given that shifts in miRNA expression profiles in response to infection have already been reported (as the authors quote), and merely cataloging differential expression (of miRNAs and their variants) gives little indication as to the functional consequence. With this in mind, my MAIN points to address is as follows...

1) The authors state that "32% of miRNAs presented a frequent shift of the 5' start site" (pg 6). A similar shift is discussed with relation to miR-146a-3p (pg 11) leading to many differential targets being predicted. In my opinion, the manuscript would be strengthened significantly by attempting to validate that an isomiR shift in response to infection brought about a change in the targeting of a process, or even specific target gene, in line with the isomiR shift. Even if this could not be shown, the attempt to demonstrate this on a reasonable scale would also tell us something quite concrete about the significance of the isomiR shifts.

Other points for consideration:

2) On pg 7 (and several times later in the text) the authors compare more minor changes in miRNAs to larger changes in protein coding genes. Could the authors clarify how similar this study is to Ref#21 where a specific comparison is made between results for miRNAs in this study and results for mRNAs in Ref#21

3) On pg 7, "340 miRNAs..presented differential expression upon stimulation". This is a lot of miRNAs. I see no reason for this to be incorrect, but wonder how many miRNAs significantly change which are very lowly expressed to begin with (and hence, are likely of little consequence). Could expression be built into such data to indicate how frequently highly expressed (and obviously functional) miRNAs are differentially regulated?

4) Related to the above comment, with so many miRNAs changing, is this based on something like changing ratios of counts per million? If so, is it possible something else is changing significantly in one condition, thus throwing out cpm / ratios for everything else? (I wouldn't think so, but worth checking).

5) On pages 13,14, it is described how few miRNAs seem to downregulate the level of their targets (at mRNA level). If one divides miRNA:targets into very highly predicted relationships down to weakly predicted strength of targeting, does it become apparent that in the highly predicted subgroup, negative expression correlation becomes more common?

6) Related to above, is it possible to consider relationships not to one specific miRNA, but for examples where 2 or more co-regulated miRNAs target the same gene. Do combinations of miRNAs show more apparent regulatory effects than when each miRNA is considered separately?

Reviewer 2

Are you able to assess all statistics in the manuscript, including the appropriateness of statistical tests used?
Yes, and I have assessed the statistics in my report.

Comments to author:

The manuscript by Rotival et al. aims to dissect the extent to which genetic variation among populations influences variation in miRNA expression levels, modifications, and miRNA responses to immune stimulation. The authors collect genotyping information, miRNA and mRNA sequencing profiles from 200 individuals of African and European ancestry, before and after activation with four different immune stimuli. With these data, they do a very nice systematic analysis of miRNA responses (and modifications), characterize population differences in miRNA expression, and identify miR-QTLs, QTLs associated with isomiRs, and the extent to which these QTLs might be underlying eQTLs in the same data. Overall, the study was well described, well controlled, and presented some interesting and valuable results. I have a few concerns that I think the authors should be able to address:

Major concerns:

- 1) I would like the authors to spend a little more time differentiating between monocyte specific or stimuli specific miRNA profiles and modifications. Specifically, how do some of the modification signature that they're seeing (3' variation, isomiR shortening, substitutions indicative of modifications) differ before and after immune stimulation?
- 2) When the authors try to assess the correlation between miRNA expression and mRNA expression levels across samples, I wasn't quite sure if this was an all by all comparison (ie. Independently correlating every miRNA expression level with every mRNA expression level). If so, this approaches seems hard to interpret since a miRNA might be correlated with a number of mRNA expression levels through influence on an entire transcriptional network. I would suggest that the authors should also do a more targeted analysis - specifically do the correlations between miRNA expression levels and mRNAs which contain targets for those miRNAs in their 3' UTRs.
- 3) How many of the miRNAs that the authors evaluate are transcribed from introns of mRNAs? For these miRNAs, it would be interesting to assess if the same QTL serves as a mRNA eQTL and a miR-QTL and if the QTL is dependently (through transcription of the mRNA) or independently associated with both expression levels.
- 4) In a few places, I think the authors are slightly overstating their conclusions to a greater extent than is warranted from their results. For instance:
 - a. On p. 11, the authors state that "genetic variants altering isomiR ratios have a significant impact on immune response variability." This conclusion seems to be based primarily on one example of rs2910164, which is a miR-QTL and associated with allergic rhinitis and asthma. Is this SNP is associated with any other molecular quantitative traits (ie. eQTLs)? It is unclear whether this SNP is directly influencing immune phenotypes through the miRNA or could be influencing any other traits as well.
 - b. On p. 15, the authors state that "our work highlights the importance of considering isomiR changes, and not only miRNA expression, when studying the impact of miRNAs on immune responses." While I do think it is molecularly interesting to start to study miRNA modifications, I don't think the authors really show any function to these isomiR changes, but only highlight that there is variability in isomiR expression that maybe (potentially randomly) associated with genetic variation.
 - c. On p. 16, the authors state that "This suggests that population differences in miRNA responses are driven by specific environmental exposures, possibly through epigenetic priming of innate immune responses." The authors are making a pretty broad statement here without really spelling out the rationale for what these non-genetic factors could be or what they mean by environmental exposures or epigenetic priming that differs between populations. Additionally, I think they might actually be too quick to rule out genetic differences between populations that may underlie differences in miRNA responses - these could be indirect genetic effects that influence upstream components or feedback loops involved in miRNA biogenesis pathways.
- 5) In the discussion, the authors spend a lot of time discussing the scant evidence that miRNAs affect mRNA expression levels (and thus mRNA stability) in their system. This seems to suggest that miRNA responses may not have very much impact at all on gene regulatory processes here. This downplays the potential that aggregate contributions of these small effects could play a large role in fine-tuning mRNA pathways (something we know very

little about, but has been hypothesized and modeling in miRNA literature). Furthermore, the authors never mention the potential impacts that miRNA expression could have on translational efficiency, which is the other potential molecular function of miRNAs.

Minor concerns:

- 1) Are miRNAs really "epigenetic" regulators? Since the point of the paper is that miRNA expression (including the change in miRNA expression after immune response) is very likely to be regulated by genetic variation, I would argue that miRNAs are not quite epigenetic.
- 2) The authors state on p. 9 that there is a "decreased proportion of miR-QTLs among highly expressed miRNAs" - could this be due to a technical bias? For instance, maybe there is a lot more variability in miRNA expression at higher expression levels, thus reducing the power to see lower effect sizes?
- 3) Similarly, the authors state on p. 10 that "miR-QTLs are largely insensitive to stimulation compared to eQTLs of protein-coding genes." Could this be due to the miRNAs themselves being less sensitive to large changes upon stimulation or that there is less power in these analyses to properly identify subtle changes in miR-QTL usage upon stimulation?

Point-by-point responses to reviewers' comments

Reviewer #1:

The manuscript is well written and comprehensively details the cataloging of miRNA sequence variants across an impressively large number of samples (monocytes from 200 individuals, treated separately with 4 stimuli). The area of isomiRs is potentially significant as miRNA sequence variation is extensive, though there remains a lack of understanding how widely regulated these differences are and what their functional consequences are likely to be.

I have several comments for the authors to consider, though my main critique is the impact of the cataloging of such variants given that shifts in miRNA expression profiles in response to infection have already been reported (as the authors quote), and merely cataloging differential expression (of miRNAs and their variants) gives little indication as to the functional consequence. With this in mind, my MAIN points to address is as follows...

1) The authors state that "32% of miRNAs presented a frequent shift of the 5' start site" (pg 6). A similar shift is discussed with relation to miR-146a-3p (pg 11) leading to many differential targets being predicted. In my opinion, the manuscript would be strengthened significantly by attempting to validate that an isomiR shift in response to infection brought about a change in the targeting of a process, or even specific target gene, in line with the isomiR shift. Even if this could not be shown, the attempt to demonstrate this on a reasonable scale would also tell us something quite concrete about the significance of the isomiR shifts.

RESPONSE: We agree with the reviewer that the question of the functional consequences of isomiR shifts is essential to evaluate the relevance of the observed differences in isomiRs ratios. To address this question, we have now performed several new analyses that explore, in a systematic way, the impact of the observed isomiR modifications on the predicted targets of miRNAs.

Specifically, we have used miRanda to comprehensively predict targets associated to frequent non-canonical isomiRs and compare them with the predicted targets of canonical isomiRs. In doing so, we report that shifts of the 5' boundary of miRNAs are predicted to alter between 43 and 93% of miRNA targets. Conversely, shifts of the 3' boundary are predicted to have only a minor effect on miRNA targets, typically affecting <0.1% of targets (**see P9 L10–26** of the new section entitled « Functional impact of miRNA modifications upon immune stimulation », as well as **Figure 3i,j and Table S2D**). Note, however, that the impact of 3' end shifts might be under-estimated, as most prediction software do not consider non-canonical binding, which is thought to be more dependent on complementarity of the 3' end (see **Discussion, P18 – L14-17**).

We also tested whether isomiR-specific targets were enriched in specific biological functions and found that, in >93% of cases, these were not targeting any specific functional category (**Figure 3k**). Yet, we report 26 miRNAs for which targets that are unique either to the canonical or to the non-canonical isomiRs are significantly enriched in specific functional categories (**Table S2E**). For instance, upon stimulation, miR-6503-3p decreases expression of its canonical isomiR in favour of isomiRs displaying non-canonical 5' start sites, three of which are predicted to bind preferentially to type I interferons genes.

We now describe these results in the new section « Functional impact of miRNA modifications upon immune stimulation » (**P9 L26 to P10 L 13** and discuss them in the Discussion section (**P18 – L14-21**). We thank the reviewer for their remark as introducing this new section in the manuscript has strengthened our conclusions.

Other points for consideration:

2) On pg 7 (and several times later in the text) the authors compare more minor changes in miRNAs to larger changes in protein coding genes. Could the authors clarify how similar this study is to Ref#21 where a specific comparison is made between results for miRNAs in this study and results for mRNAs in Ref#21

RESPONSE: We thank the reviewer for pointing this out, as it shows that the link between this study and Ref#21 was unclear. Indeed, both studies are based on the same individuals. After stimulation, RNA was extracted from CD14+ monocytes and 2 aliquots were used to extract, on the one hand, mRNAs that served for Ref#21 (and that we re-use in the present study for comparison) and, on the other hand, small RNAs that are being published and analysed in this study for the first time. Therefore, mRNAs (from Ref#21) and miRNAs from this study are perfectly comparable as they were extracted from the same individuals and samples. To clarify this, we now state that « the effect of immune activation on miRNA expression is much weaker than that observed for protein-coding mRNAs measured on the same individual samples [21], » (see **P7 L21-22**).

3) On pg 7, "340 miRNAs..presented differential expression upon stimulation". This is a lot of miRNAs. I see no reason for this to be incorrect, but wonder how many miRNAs significantly change which are very lowly expressed to begin with (and hence, are likely of little consequence). Could expression be built into such data to indicate how frequently highly expressed (and obviously functional) miRNAs are differentially regulated?

RESPONSE: To address this question, we have now assessed how many miRNAs are differentially expressed (DE) between conditions as a function of steady-state gene expression. We find that DE miRNAs are observed, interestingly, across all levels on gene expression, with a slightly higher frequency among genes with >10 CPM (OR=2.0, Fisher's p < 4.4×10^{-16}). These results have been added to the manuscript and in a new figure panel (see **P8 L1-3, and Figure S3a**).

4) Related to the above comment, with so many miRNAs changing, is this based on something like changing ratios of counts per million? If so, is it possible something else is changing significantly in one condition, thus throwing out cpm / ratios for everything else? (I wouldn't think so, but worth checking).

RESPONSE: The differential expression analysis is indeed based on counts per millions of miRNA-aligned reads. These counts are normalised for library size using SizeFactors estimated by DESeq2, which subtracts from each sample the median difference between log counts of the samples and the average log count across all samples. As a result, we wouldn't expect a systematic shift in log counts between conditions, even if a single miRNA displays extreme variation upon stimulation.

To confirm this, we have now visualized the distribution of log fold changes across various levels of (basal) miRNA expression. If the effect was driven by one or a few miRNAs changing drastically, we would expect a handful of highly expressed miRNAs to display extreme variation in one direction while the remaining miRNAs should be homogeneously shifted in the opposite direction. Yet, we see a balanced distribution of up- and down-regulated miRNAs across all levels of gene expression, at odds with the hypothesis of a single cause underlying variation in miRNA levels upon stimulation. This new visualization has been added as **Figure S3b**. In addition, we now explicitly detail the principle of SizeFactors normalization in the **Methods section (P23, L13-16)**.

5) On pages 13,14, it is described how few miRNAs seem to downregulate the level of their targets (at mRNA level). If one divides miRNA:targets into very highly predicted relationships

down to weakly predicted strength of targeting, does it become apparent that in the highly predicted subgroup, negative expression correlation becomes more common?

RESPONSE: We thank the reviewer for this suggestion. Unfortunately, miRanda does not allow to distinguish between highly-predicted and weakly predicted targets. However, we have now compared targets predicted by miRanda with those predicted using 4 different algorithms (targetScan, mirTarget, MIRZA & MIRZA-G). We have then counted the number of algorithms where an interaction was predicted, and used this number as a measure of the ‘confidence’ in the miRNA-gene interactions. We found no enrichment of significant or negative correlations among high-confidence targets, regardless of the degree of confidence in the miRNA-gene interaction. These results are now reported in **P15 L25 to P16 L2 and Figure S5c,d**).

Figure 1: Lack of enrichment for negative correlation among highly predicted targets from targetScan (upper panels) and mirDB (lower panels). (a,d) targetScan context scores and mirTargets binding scores correlate with the number of databases where the miRNA-gene interaction is predicted. (b,e) Lack of enrichment for negative correlations with increasing scores for miRNA-gene interaction. (thresholds: targetScan - weak [0-0.2], moderate [0.2-0.8], strong [0.8,1], mirTarget: weak [0.5-0.6], moderate [0.6-0.9], strong [0.9-1]). (c,f) Lack of enrichment for stronger correlation with increasing scores for miRNA-gene interaction, same threshold as b,e.

Similar results were found when considering targetScan’s context scores or mirTargets binding scores, both of which show strong correlation with our confidence metric based on the sharing of miRNA targets (**Figure 1** for the reviewer).

6) Related to above, is it possible to consider relationships not to one specific miRNA, but for examples where 2 or more co-regulated miRNAs target the same gene. Do combinations of miRNAs show more apparent regulatory effects than when each miRNA is considered separately?

RESPONSE: We thank the reviewer for this interesting suggestion. To test whether combinations of cooperating miRNAs may interact to regulate gene expression, we first searched for pairs of miRNAs whose binding sites co-localize more often than expected by chance within 20bp of each other, in the 3’UTR regions. We then focused on 390 such miRNA pairs with adjacent binding sites in >100 expressed genes. We then searched for combined effects of the miRNA pair on gene expression, allowing for both additive and multiplicative effects, and compared the effects of miRNAs pair between genes with a pair of binding sites in their 3’ UTR and the rest of the genome.

In doing so, we found that correlation of pairs of interacting miRNAs with their shared targets did not exceed the correlation observed with the rest of the genome, supporting further the notion of a limited contribution of miRNAs to inter-individual variability, owing to strong selective constraints (see **P16 L3-11 and Figure S5e**). Furthermore, we now discuss this topic more extensively in the Discussion section to clarify that « [the limited impact of miRNAs on population differences in mRNA expression levels] could be explained both by small effects of individual miRNAs on gene expression and by a reduced variability of miRNA expression, as suggested by the limited genetic control of miRNA expression. Yet, this does not preclude an important role of miRNAs in the regulation of gene expression through the aggregate contribution of a large number of miRNAs, or in the fine-tuning of protein responses through translational inhibition. (see **Discussion P20 L6-13**).

Reviewer #2

The manuscript by Rotival et al. aims to dissect the extent to which genetic variation among populations influences variation in miRNA expression levels, modifications, and miRNA responses to immune stimulation. The authors collect genotyping information, miRNA and mRNA sequencing profiles from 200 individuals of African and European ancestry, before and after activation with four different immune stimuli. With these data, they do a very nice systematic analysis of miRNA responses (and modifications), characterize population differences in miRNA expression, and identify miR-QTLs, QTLs associated with isomiRs, and the extent to which these QTLs might be underlying eQTLs in the same data. Overall, the study was well described, well controlled, and presented some interesting and valuable results. I have a few concerns that I think the authors should be able to address:

Major concerns:

1) I would like the authors to spend a little more time differentiating between monocyte specific or stimuli specific miRNA profiles and modifications. Specifically, how do some of the modification signature that they're seeing (3' variation, isomiR shortening, substitutions indicative of modifications) differ before and after immune stimulation?

RESPONSE: We agree that this is an important question. The rationale for presenting modifications signatures jointly for basal and stimulated states was that the overall profiles of miRNA modifications (i.e. the frequency of 3' variation, isomiR shortening, and substitutions) are largely unchanged by stimulation, as illustrated by the distribution of miRNA modifications in non-stimulated and IAV-stimulated samples (**Figure 2** for the reviewer). Following the reviewer's suggestion, we now clarify this point and state that « We found that the overall frequency of miRNA modifications was virtually unchanged by stimulation (Wilcoxon $p > 0.05$, for all stimuli relative to basal state), with shifts in 3' end site of miRNAs being the most frequent type of modification » (see **P6 L6-9**)

Figure 2 : Compared distribution of miRNA modifications between non-stimulated (upper panels) and IAV conditions (lower panels). For each possible miRNA modification, we report the percentage of miRNAs for which the corresponding isomiRs accounts for at least 1%, 5%, 10% and 50% of edited reads.

In addition, note that although the overall distribution of miRNA modifications is unchanged upon stimulation, pairwise comparisons reveal weak (<1% change in isomiR ratio) but consistent shifts in frequency of isomiR modifications upon stimulation. Such shifts are discussed (**P8 L26 to P9 L6**) and characterized in **Table S2C**, and we have clarified the relation of these analyses to the comparison of the overall profile of miR modifications.

2) When the authors try to assess the correlation between miRNA expression and mRNA expression levels across samples, I wasn't quite sure if this was an all by all comparison (ie. Independently correlating every miRNA expression level with every mRNA expression level). If so, this approaches seems hard to interpret since a miRNA might be correlated with a number of mRNA expression levels through influence on an entire transcriptional network. I would suggest that the authors should also do a more targeted analysis - specifically do the correlations between miRNA expression levels and mRNAs which contain targets for those miRNAs in their 3' UTRs.

RESPONSE: We thank the reviewer for raising this point. When assessing correlation between miRNAs and gene expression, we indeed performed an “all-by-all” comparison where we measured, for each gene, the predictive power of each individual miRNA (adjusted for all other miRNAs). We entirely agree with the reviewer that mRNA expression might be correlated to a miRNA through the influence on the entire transcriptional network. This is exactly the reason why we believe that this justifies the approach of doing an all-by-all comparison and compare targets to non-targets, rather than focusing on predicted targets, to increase power for the detection of miRNA-gene correlations. Indeed, because spurious correlations can affect equally targets and non-targets, it is necessary to use non-targets as controls to assess the relevance of the correlations that are observed among predicted targets.

However, following the reviewer’s question, we have now performed additional analysis, where we specifically test the downstream effect of miR-QTLs on predicted targets of their associated miRNAs (**presented in P17 L12-14 and Figure S5f**), thus increasing power to detect associations. Our conclusions remain unchanged with this new analysis. Finally, because some readers might be interested in this topic, we now provide as a supplementary table (**Table S5**) the significant associations between miRNAs and their target genes, across all 5 experimental conditions.

3) How many of the miRNAs that the authors evaluate are transcribed from introns of mRNAs? For these miRNAs, it would be interesting to assess if the same QTL serves as a mRNA eQTL and a miR-QTL and if the QTL is dependently (through transcription of the mRNA) or independently associated with both expression levels.

RESPONSE: We thank the reviewer for suggesting this interesting analysis. In total, 352 of the miRNAs that we identified overlap with a gene that is deemed as expressed in our data (FPKM>1 in at least one condition). Of these genes, 81 have an eQTL 20 of which display significant associations with the miRNA they carry. Yet, of the 64 miR-QTLs that affect intronic miRNAs, only 26 have an eQTL at their target gene, and 6 are in high LD with the eQTL of their host gene. This indicates that genetic control of miRNA expression is largely independent from that of their host genes. Overall, we find only one case (miR-147b) where the miR-QTL is consistent with a direct regulation of the miRNA through regulation of the host gene. These new results have been now added in the manuscript (**P12, L15-26 of the new section entitled « Genetic control of miRNAs is largely independent from that of protein-coding genes »**).

4) In a few places, I think the authors are slightly overstating their conclusions to a greater extent than is warranted from their results. For instance:

a. On p. 11, the authors state that "genetic variants altering isomiR ratios have a significant impact on immune response variability." This conclusion seems to be based primarily on one example of rs2910164, which is a miR-QTL and associated with allergic rhinitis and asthma. Is this SNP is associated with any other molecular quantitative traits (ie. eQTLs)? It is unclear whether this SNP is directly influencing immune phenotypes through the miRNA or could be influencing any other traits as well.

RESPONSE: We agree with the reviewer that there is a possibility that some variants could alter disease risk through their effect on other genes at the locus (**Figure 3** for the reviewer). In the GTEx dataset, however, no genome-wide significant association was found for this specific variant. In the mRNA data from ref. [1], which is based on the same samples, the strongest association of rs2910164 was with

Figure 3: Genes present at the rs2910164 locus. Vertical red line highlights the position of the SNP. Protein coding genes are shown in yellow and non-coding RNAs in purple. Uncharacterized transcripts are in grey.

ATP10B ($P < 9 \times 10^{-7}$), a phospholipid transporter involved in susceptibility to Parkinson's disease, with no known function in immunity. Thus, given the known role of miR-146a in innate immunity [2], the effect of rs2910164 on miR-146a isomiRs seems the most plausible mechanism underlying allergic rhinitis and asthma susceptibility at the locus. Yet, to avoid any overstatement, we have rephrased the sentence accordingly to: « these results highlight how genetic variants altering isomiR ratios can lead to a profound rewiring of targets from key immune regulators » (P13 L15-17)

b. On p. 15, the authors state that "our work highlights the importance of considering isomiR changes, and not only miRNA expression, when studying the impact of miRNAs on immune responses." While I do think it is molecularly interesting to start to study miRNA modifications, I don't think the authors really show any function to these isomiR changes, but only highlight that there is variability in isomiR expression that maybe (potentially randomly) associated with genetic variation.

RESPONSE: We thank the reviewer for this remark. Following this comment, as well as that of reviewer 1, we now assess in a systematic way the consequences of isomiR changes on predicted targets (see P9 L10 to P10 L13 of the new section entitled « Functional impact of miRNA modifications upon immune stimulation » as well as Figure 3i-k).

Briefly, while we find that a majority of isomiR changes have little impact on predicted targets, we report that >19% of non-canonical isomiRs are associated with a change in their targets, this proportion decreasing among highly expressed isomiRs. We further find that for 6.8% of the tested miRNAs, modifications of isomiR expression are associated with non-random gains or losses of targets, which are biased toward specific biological functions. Among these, we find 10 miRNAs that change their isomiR levels upon stimulation. For example, our analysis predicts that shift in 5' start miR-449c-5p targets is associated with a loss of targets involved in cell-cell adhesion, consistent with the difference in effect of the 5' shifted and canonical isomiRs on *in vitro* cell adhesion phenotypes [3]. We also report an effect of 5' isomiR of miR-6503-3p that are up-regulated following stimulation by viral cues and preferentially target type-I IFNs, supporting a functional impact of isomiR changes observed in response to infection. Overall, we believe that these updated results better highlight how isomiR changes may indeed impact immune responses.

c. On p. 16, the authors state that "This suggests that population differences in miRNA responses are driven by specific environmental exposures, possibly through epigenetic priming of innate immune responses." The authors are making a pretty broad statement here without really spelling out the rationale for what these non-genetic factors could be or what they mean by environmental exposures or epigenetic priming that differs between populations. Additionally, I think they might actually be too quick to rule out genetic differences between populations that may underlie differences in miRNA responses - these could be indirect genetic effects that influence upstream components or feedback loops involved in miRNA biogenesis pathways.

RESPONSE: We entirely agree with the reviewer in that indirect genetic effects could indeed contribute to the observed population differences in miRNA expression. We have now completely rewritten this part of the Discussion section and mention that: « The limited genetic control of miRNA expression, with respect to protein-coding genes, indicates that most of the observed population differences are attributable either to *trans*-acting genetic factors regulating the miRNA biogenesis/decay pathways or to non-genetic factors » (see P19 L7-10). These non-genetic factors could be indeed related to socio-economic environment, diet, behavioral features and/or previous pathogen exposure.

In addition, we have rephrased our statement on «epigenetic priming of innate immune responses » to make more apparent that (i) miRNAs that respond differently between

populations consistently display a stronger response in Africans, and (ii) given the established role of these miRNAs in induction of LPS tolerance, this trend could indicate a « weaker innate immunity response to secondary challenges among African-descent individuals » (see P19 L12-17).

5) In the discussion, the authors spend a lot of time discussing the scant evidence that miRNAs affect mRNA expression levels (and thus mRNA stability) in their system. This seems to suggest that miRNA responses may not have very much impact at all on gene regulatory processes here. This downplays the potential that aggregate contributions of these small effects could play a large role in fine-tuning mRNA pathways (something we know very little about, but has been hypothesized and modeling in miRNA literature). Furthermore, the authors never mention the potential impacts that miRNA expression could have on translational efficiency, which is the other potential molecular function of miRNAs.

RESPONSE: We thank the reviewer for this remark, which has helped us clarifying our statement. Accordingly, we have rephrased the Discussion as follows: « [...] the lack of measurable effects of miR-QTLs on gene expression suggests that individual miRNAs have only a limited impact on population differences of mRNA expression levels. This could be explained both by small effects of individual miRNAs on gene expression and by a reduced variability of miRNA expression, as suggested by the limited genetic control of miRNA expression. Yet, this does not preclude an important role of miRNAs in the regulation of gene expression through the aggregate contribution of a large number of miRNAs, or in the fine-tuning of protein responses through translational inhibition» (see Discussion section P20, L6-13).

Minor concerns:

1) Are miRNAs really "epigenetic" regulators? Since the point of the paper is that miRNA expression (including the change in miRNA expression after immune response) is very likely to be regulated by genetic variation, I would argue that miRNAs are not quite epigenetic.

RESPONSE: We have now accordingly removed the mention of miRNAs as epigenetic regulators, as this terminology is controversial.

2) The authors state on p. 9 that there is a "decreased proportion of miR-QTLs among highly expressed miRNAs" - could this be due to a technical bias? For instance, maybe there is a lot more variability in miRNA expression at higher expression levels, thus reducing the power to see lower effect sizes?

RESPONSE: We agree with the reviewer that a higher technical variability among highly expressed miRNAs would lead to an apparent reduction of miR-QTLs. Yet, we have several reasons to believe that the reduction in miR-QTLs that we observe is not driven by a technical factor. First, while we expect to see an increase of technical variability with average gene/miRNA expression, standard transcriptomics pipelines include a variance stabilizing transform (in our study, a log₂ transform with an offset of 1) that removes this dependency by transforming multiplicative errors (that increase their variance with gene expression) into additive errors (with a constant variance across all bins of expression, see Figure 4 for the reviewer). Second, the effectiveness of our variance-

Figure 4 : Relationship between miRNA expression and variance (standard deviation), before (a) and after log transformation (b)

stabilizing transform (in our study, a log₂ transform with an offset of 1) that removes this dependency by transforming multiplicative errors (that increase their variance with gene expression) into additive errors (with a constant variance across all bins of expression, see Figure 4 for the reviewer). Second, the effectiveness of our variance-

stabilizing transformation can be validated empirically by comparing the trend of decreasing number of miR-QTLs with higher expression, with that observed for eQTLs of protein-coding genes. Indeed, the same trend of stronger technical variability among highly expressed genes is expected for the mRNA data. Yet, we don't see a clear decrease of the number of eQTLs with higher gene expression (see **Figure S4a**), which supports that our findings are not driven by higher technical variation among highly expressed miRNAs. Third, if there remained a higher technical variability among highly expressed miRNAs after log-transforming the data, we would expect this to also impact the detection of differential expression following stimulation. Yet, when looking at the proportion of miRNAs that are differentially expressed upon stimulation, we do not see such a steady decrease with the miRNA expression (see **Figure S3a**). Altogether, this suggests that the decreased proportion of miR-QTLs among highly expressed miRNAs is not driven by increased technical noise.

3) Similarly, the authors state on p. 10 that "miR-QTLs are largely insensitive to stimulation compared to eQTLs of protein-coding genes." Could this be due to the miRNAs themselves being less sensitive to large changes upon stimulation or that there is less power in these analyses to properly identify subtle changes in miR-QTL usage upon stimulation?

RESPONSE: We thank the reviewer for raising this point. We entirely agree with the reviewer that the lower sensitivity of miR-QTLs to stimulation, compared to eQTLs, could result from the lower sensitivity of miRNA expression to stimulation. To test this, we first quantified the maximum fold change of miRNA and protein-coding genes upon stimulation. We found that while 99% of protein-coding genes display an absolute log₂ fold change > 0.2 for at least one stimulus, only 18% of miRNAs display a similar response upon stimulation. We then used logistic regression to model the robustness of eQTLs and miR-QTLs to stimulation, as a function of their absolute log₂ fold change (considering the maximum fold change across stimuli) and tested whether, for a similar fold change, miR-QTLs were more or less likely to be robust to stimulation compared to eQTLs. In doing so, we found that while the robustness of molecular QTLs is generally lower among molecular traits that respond the most to stimulation, this effect does not suffice to account for the increased stability of miR-QTLs across conditions. This new analysis has now been added in the manuscript (see **P11 L14-20**).

Regarding the second point, we believe that lack of power is unlikely to cause the apparent robustness of miR-QTLs to stimulation. For each miR-QTL and eQTL, we compare the likelihood for various models of sharing across conditions, and select the most likely model. While an increase in noise could indeed prevent the detection of significant GxE interactions, it should have only a limited impact on our likelihood-based model selection. If anything, an increased noise in miRNA data would lead to select a random model, resulting in a low probability to predict a shared effect across all conditions. In light of this, we remain confident that the lower number of condition-dependent miR-QTLs is not an artefact due to increase noise in the miRNA data.

References:

1. Quach, H., et al., *Genetic Adaptation and Neandertal Admixture Shaped the Immune System of Human Populations*. Cell, 2016. **167**(3): p. 643-656 e17.
2. Su, Y.L., et al., *Myeloid cell-targeted miR-146a mimic inhibits NF-kappaB-driven inflammation and leukemia progression in vivo*. Blood, 2020. **135**(3): p. 167-180.
3. Mercey, O., et al., *Characterizing isomiR variants within the microRNA-34/449 family*. FEBS letters, 2017. **591**(5): p. 693-705.

Second round of review

Reviewer 1

A thorough response to reviewer comments were made, including new analyses which have further extended and strengthened the manuscript. I accept that this is an in silico analysis and although the impact of isomiR shifts would have been greatly enhanced by wet bench validation, it is also a very large task to do this at the sort of scale required to drive firm conclusions.

Reviewer 2

The authors have satisfactorily addressed all of my comments on the manuscript and I feel it is now suitable for publication in Genome Biology.